# CAMBranch: Contrastive Learning with Augmented MILPs for Branching

**Jiacheng Lin**[1,†*]    **Meng Xu**[2†]    **Zhihua Xiong**[2]    **Huangang Wang**[2‡]

[1]University of Illinois Urbana-Champaign    [2]Tsinghua University

## Abstract

Recent advancements have introduced machine learning frameworks to enhance the Branch and Bound (B&B) branching policies for solving Mixed Integer Linear Programming (MILP). These methods, primarily relying on imitation learning of Strong Branching, have shown superior performance. However, collecting expert samples for imitation learning, particularly for Strong Branching, is a time-consuming endeavor. To address this challenge, we propose **C**ontrastive Learning with **A**ugmented **M**ILPs for **Branch**ing (CAMBranch), a framework that generates Augmented MILPs (AMILPs) by applying variable shifting to limited expert data from their original MILPs. This approach enables the acquisition of a considerable number of labeled expert samples. CAMBranch leverages both MILPs and AMILPs for imitation learning and employs contrastive learning to enhance the model's ability to capture MILP features, thereby improving the quality of branching decisions. Experimental results demonstrate that CAMBranch, trained with only 10% of the complete dataset, exhibits superior performance. Ablation studies further validate the effectiveness of our method.

## 1 Introduction

Mixed Integer Linear Programming (MILP) is a versatile tool for solving combinatorial optimization problems, with applications across various fields (Bao & Wang, 2017; Soylu et al., 2006; Godart et al., 2018; Almeida et al., 2006; Hait & Artigues, 2011). A prominent approach for solving MILPs is the Branch-and-Bound (B&B) algorithm (Land & Doig, 1960). This algorithm adopts a divide-and-conquer approach, iteratively resolving sub-problems and progressively reducing the search space. Within the execution of the algorithm, one pivotal decision comes to the fore: variable selection, also known as branching. Traditionally, variable selection relies heavily on expert-crafted rules rooted in substantial domain knowledge. However, recent developments have seen a shift of focus towards the integration of machine learning based frameworks, aiming to enhance the B&B algorithm by replacing conventional, hand-coded heuristics (Gasse et al., 2019; Zarpellon et al., 2021; Nair et al., 2020; Lin et al., 2022). This transition marks a notable advancement in the field, leveraging the power of machine learning and data-driven approaches to tackle complex problems more effectively. For a comprehensive overview of the notable developments in this emerging field, refer to the survey provided in Bengio et al. (2021).

The performance of the B&B algorithm hinges significantly upon its branching strategy, and sub-optimal branching decisions can exponentially escalate the computational workload. This predicament attracts researchers to explore the integration of machine learning (ML) techniques to enhance branching strategies. Notably, Gasse et al. (2019) have trained branching policy models using imitation learning, specifically targeting Strong Branching (Applegate et al., 1995), a traditional strategy known for generating minimal branching search trees but with extremely low efficiency. By mapping a MILP into a bipartite, these branching policy models leverage Graph Convolution Neural Networks (GCNN) (Kipf & Welling, 2017) to extract variable features and make variable selec-

---

*This work was completed during the master program at Tsinghua University.

†Equal contributions.

‡Corresponding to `hgwang@tsinghua.edu.cn`.

tion decisions. This approach has demonstrated superior performance in solving MILPs, marking a significant milestone in the application of machine learning to MILP solving.

Despite making significant progress, a significant challenge arises with the imitation learning paradigm mentioned above. The collection of expert samples for imitation learning requires solving numerous MILP instances using Strong Branching, which is computationally intensive and time-consuming. From our experiments, collecting 100k expert samples for four combinatorial optimization problems (*Easy* level) evaluated in (Gasse et al., 2019), namely the Set Covering Problem (BALAS, 1980), Combinatorial Auction Problem (Leyton-Brown et al., 2000), Capacitated Facility Location Problem (Cornuejols et al., 1991), and Maximum Independent Set Problem (Cire & Augusto, 2015), took a substantial amount of time: 26.65 hours, 12.48 hours, 84.79 hours, and 53.45 hours, respectively. These results underscore the considerable effort and resources required for collecting a sufficient number of expert policy samples even on the *Easy* level instances. Importantly, as the complexity of MILPs scales up, the challenge of collecting an adequate number of samples for imitation learning within a reasonable timeframe becomes increasingly impractical.

To address this issue, we present a novel framework named **C**ontrastive Learning with **A**ugmented **M**ILPs for **Branch**ing (CAMBranch). Our approach begins with the development of a data augmentation technique for MILPs. This technique generates a set of Augmented MILPs (AMILPs) through variable shifting, wherein random shifts are applied to each variable within a MILP to produce a new instance. This augmentation strategy enables the acquisition of a substantial number of labeled expert samples, even when expert data is limited. It eliminates the need for extensive computational efforts associated with solving numerous MILP instances, thereby mitigating the challenges related to expert strategy sample collection. Next, building upon the work of Gasse et al. (2019), we transform a MILP into a bipartite graph. By providing theoretical foundations and proofs, we establish a clear correspondence between an augmented bipartite graph (derived from an AMILP) and its corresponding original bipartite graph. These bipartite graph representations are then fed into Graph Convolutional Neural Networks (GCNNs) to extract essential features and make branching decisions. Finally, we employ contrastive learning between MILPs and corresponding AMILPs to facilitate policy network imitation learning. This choice is motivated by the fact that MILPs and their AMILP counterparts share identical branching decisions, enabling a seamless integration of this learning approach. We evaluate our approach on four classical NP-hard combinatorial optimization problems, following the experimental setup described in Gasse et al. (2019). The experimental results demonstrate the superior performance of our proposed CAMBranch, even if CAMBranch leverages only 10% of the data used in Gasse et al. (2019).

## 2 PRELIMINARIES

### 2.1 MIXED INTEGER LINEAR PROGRAMMING (MILP)

The general definition form of a MILP problem instance $\text{MILP} = (\boldsymbol{c}, \boldsymbol{A}, \boldsymbol{b}, \boldsymbol{l}, \boldsymbol{u}, \boldsymbol{\mathcal{I}})$ is shown below

$$\min_{\boldsymbol{x}} \ \boldsymbol{c}^{\mathrm{T}}\boldsymbol{x} \quad \text{s.t. } \boldsymbol{Ax} \leqslant \boldsymbol{b}, \ \boldsymbol{l} \leqslant \boldsymbol{x} \leqslant \boldsymbol{u}, \ x_j \in \mathbb{Z}, \ \forall j \in \boldsymbol{\mathcal{I}} \tag{1}$$

where $\boldsymbol{A} \in \mathbb{R}^{m \times n}$ is the constraint coefficient matrix in the constraint, $\boldsymbol{c} \in \mathbb{R}^n$ is the objective function coefficient vector, $\boldsymbol{b} \in \mathbb{R}^m$ represents the constraint right-hand side vector, while $\boldsymbol{l} \in (\mathbb{R} \cup \{-\infty\})^n$ and $\boldsymbol{u} \in (\mathbb{R} \cup \{+\infty\})^n$ represent the lower and upper bound vectors for each variable, respectively. The set $\boldsymbol{\mathcal{I}}$ is an integer set containing the indices of all integer variables.

In the realm of solving MILPs, the B&B algorithm serves as the cornerstone of contemporary optimization solvers. Within the framework of the B&B algorithm, the process involves branching, which entails a systematic division of the feasible solution space into progressively smaller subsets. Simultaneously, bounding occurs, which aims to establish either lower-bound or upper-bound targets for solutions within these subsets. Lower bounds are calculated through linear programming (LP) relaxations, while upper bounds are derived from feasible solutions to MILPs.

### 2.2 MILP BIPARTITE GRAPH ENCODING

Following Gasse et al. (2019), we model the MILP corresponding to each node in the B&B tree with a bipartite graph, denoted by $(\mathcal{G}, \mathbf{C}, \mathbf{E}, \mathbf{V})$, where: (1) $\mathcal{G}$ represents the structure of the bipartite

Table 1: An overview of the features for constraints, edges, and variables in the bipartite graph $\mathbf{s}_t = (\mathcal{G}, \mathbf{C}, \mathbf{E}, \mathbf{V})$ following Gasse et al. (2019).

| Type | Feature | Description |
|---|---|---|
| **C** | obj_cos_sim | Cosine similarity between constraint coefficients and objective function coefficients. |
| | bias | Normalized right deviation term using constraint coefficients. |
| | is_tight | Indicator of tightness in the linear programming (LP) solution. |
| | dualsol_val | Normalized value of the dual solution. |
| | age | LP age, which refers to the number of solver iterations performed on the LP relaxation problem without finding a new integer solution. |
| **E** | coef | Normalized constraint coefficient for each constraint. |
| **V** | type | One-hot encoding representing the type (binary variables, integer variables, implicit integer variables, and continuous variables). |
| | coef | Normalized objective function coefficients. |
| | has_lb /_ub | Indicator for the lower/upper bound. |
| | sol_is_at_lb /_ub | The lower/upper bound is equal to the solution value. |
| | sol_frac | Fractionality of the solution value. |
| | basis_status | The state of variables in the simplex base is encoded using one-hot encodin (lower, basic, upper, zero). |
| | reduced_cost | Normalized reduced cost. |
| | age | Normalized LP age. |
| | sol_val | Value of the solution. |
| | inc_val /avg_inc_val | Value/Average value in the incumbent solutions. |

graph, that is, if the variable $i$ exists in the constraint $j$, then an edge $(i, j) \in \mathcal{E}$ is connected between node $i$ and $j$ within the bipartite graph. $\mathcal{E}$ represents the set of edges within the bipartite graph. (2) $\mathbf{C} \in \mathbb{R}^{|\mathbf{C}| \times d_1}$ stands for the features of the constraint nodes, with $|\mathbf{C}|$ denoting the number of constraint nodes, and $d_1$ representing the dimension of their features. (3) $\mathbf{V} \in \mathbb{R}^{|\mathbf{V}| \times d_2}$ refers to the features of the variable nodes, with $|\mathbf{V}|$ as the count of variable nodes, and $d_2$ as the dimension of their features. (4) $\mathbf{E} \in \mathbb{R}^{|\mathbf{C}| \times |\mathbf{V}| \times d_3}$ represents the features of the edges, with $d_3$ denoting the dimension of edge features. Details regarding these features can be found in Table 1.

Next, the input $\mathbf{s}_t = (\mathcal{G}, \mathbf{C}, \mathbf{E}, \mathbf{V})$ is then fed into GCNN which includes a bipartite graph convolutional layer. The bipartite graph convolution process involves information propagation from variable nodes to constraint nodes, and the constraint node features are updated by combining with variable node features. Similarly, the variable node updates its features by combining with constraint node features. For $\forall i \in \mathcal{C}, j \in \mathcal{V}$, the process of message passing can be represented as

$$\mathbf{c}_i' = \mathbf{f}_\mathcal{C} \left( \mathbf{c}_i, \sum_j^{(i,j) \in \mathcal{E}} \mathbf{g}_\mathcal{C}(\mathbf{c}_i, \mathbf{v}_j, \mathbf{e}_{i,j}) \right) \qquad \mathbf{v}_j' = \mathbf{f}_\mathcal{V} \left( \mathbf{v}_j, \sum_i^{(i,j) \in \mathcal{E}} \mathbf{g}_\mathcal{V}(\mathbf{c}_i, \mathbf{v}_i, \mathbf{e}_{i,j}) \right) \qquad (2)$$

where $\mathbf{f}_\mathcal{C}$, $\mathbf{f}_\mathcal{V}$, $\mathbf{g}_\mathcal{C}$, and $\mathbf{g}_\mathcal{V}$ are Multi-Layer Perceptron (MLP) (Orbach, 1962) models with two activation layers that use the ReLU function (Agarap, 2018). After performing message passing (Gilmer et al., 2017), a bipartite graph with the same topology is obtained, where the feature values of variable nodes and constraint nodes have been updated. Subsequently, an MLP layer is used to score the variable nodes, and a masked softmax operation is applied to obtain the probability distribution of each variable being selected. The process mentioned above can be expressed as $\boldsymbol{P} = \text{softmax}(\text{MLP}(\mathbf{v}))$. Here, $\boldsymbol{P}$ is the probability distribution of the output variables. During the training phase, GCNN learns to imitate the Strong Branching strategy. Upon completion of the training process, the model becomes ready for solving MILPs.

## 3 METHODOLOGY

As previously mentioned, acquiring Strong Branching expert samples for imitation learning poses a non-trivial challenge. In this section, we introduce CAMBranch, a novel approach designed to address this issue. Our first step involves the generation of Augmented MILPs (AMILPs) labeled with

Strong Branching decisions, derived directly from the original MILPs. This augmentation process equips us with multiple expert samples essential for imitation learning, even when confronted with limited expert data. Subsequently, building upon the AMILPs, we proceed to create their augmented bipartite graphs. Finally, since MILPs and corresponding AMILPs share branching decisions, we view them as positive pairs. Leveraging the power of contrastive learning, we train our model to boost performance.

## 3.1 AUGMENTED MIXED INTEGER LINEAR PROGRAMMING (AMILP)

To obtain AMILPs, we adopt variable shift from $\boldsymbol{x}$ defined in Eq.(1) to $\hat{\boldsymbol{x}}$ using a shift vector $\boldsymbol{s}$, denoted as $\hat{\boldsymbol{x}} = \boldsymbol{x} + \boldsymbol{s}$, where $\boldsymbol{s}$ is a shift vector. Note that if $x_i \in \mathbb{Z}$, then $s_i \in \mathbb{Z}$; otherwise $s_i \in \mathbb{R}$. Based on this translation, we can derive a MILP from Eq.(1). To bring this model into standard form, we redefine the parameters as follows: $\hat{\boldsymbol{b}} = \boldsymbol{A}\boldsymbol{s} + \boldsymbol{b}$, $\hat{\boldsymbol{l}} = \boldsymbol{l} + \boldsymbol{s}$, and $\hat{\boldsymbol{u}} = \boldsymbol{u} + \boldsymbol{s}$. Consequently, the final expression for the AMILP model is represented as:

$$\min_{\hat{\boldsymbol{x}}} \ \boldsymbol{c}^{\mathrm{T}}\hat{\boldsymbol{x}} - \boldsymbol{c}^{\mathrm{T}}\boldsymbol{s} \quad \text{s.t.} \ \boldsymbol{A}\hat{\boldsymbol{x}} \leq \hat{\boldsymbol{b}}, \hat{\boldsymbol{l}} \leq \hat{\boldsymbol{x}} \leq \hat{\boldsymbol{u}}, \ \hat{x}_j \in \mathbb{Z}, \ \forall j \in \mathcal{I} \tag{3}$$

Through this data augmentation technique, a single MILP has the capacity to generate multiple AMILPs. It's worth noting that each MILP, along with its corresponding AMILPs, share identical variable selection decisions of the Strong Branching. We next present a theorem to demonstrate this characteristic. To illustrate this distinctive characteristic, we begin by introducing a lemma that elucidates the relationship between MILPs and AMILPs.

**Lemma 3.1.** *For MILPs Eq.(1) and their corresponding AMILPs Eq.(3), let the optimal solutions of the LP relaxation be denoted as $\boldsymbol{x}^*$ and $\hat{\boldsymbol{x}}^*$, respectively. A direct correspondence exists between these solutions, demonstrating that $\hat{\boldsymbol{x}}^* = \boldsymbol{x}^* + \boldsymbol{s}$.*

The proof of this lemma is provided in the Appendix. Building upon this lemma, we can initially establish the relationship between the optimal values of a MILP and its corresponding AMILP, denoted as $\boldsymbol{c}^{\mathrm{T}}\boldsymbol{x}^* = \boldsymbol{c}^{\mathrm{T}}\hat{\boldsymbol{x}}^* - \boldsymbol{c}^{\mathrm{T}}\boldsymbol{s}$. This equation signifies the equivalence of their optimal values. With the above information in mind, we proceed to introduce the following theorem.

**Theorem 3.1.** *Suppose that an AMILP instance is derived by shifting variables from its original MILP. When employing Strong Branching to solve these instances, it becomes evident that both the MILP and AMILP consistently produce identical variable selection decisions at each branching step within B&B.*

*Proof.* In the context of solving a MILP $\mathcal{M}$ using Strong Branching, the process involves pre-branching all candidate variables at each branching step, resulting in sub-MILPs. Solving the linear programing (LP) relaxations of these sub-MILPs provides the optimal values, which act as potential lower bounds for $\mathcal{M}$. The Strong Branching strategy chooses the candidate variable that offers the most substantial lower bound improvement as the branching variable for that step. Thus, the goal of this proof is to demonstrate that the lower bound increments after each branching step are equal when applying Strong Branching to solve both a MILP and its corresponding AMILP.

Given a MILP's branching variable $x_i$ and its corresponding shifted variable of AMILP $\hat{x}_i = x_i + s_i$, we perform branching operations on both variables. Firstly, we branch on $x_i$ to produce two subproblems for MILP, which are formulated as follows:

$$\underset{\boldsymbol{x}}{\arg\min} \left\{ \boldsymbol{c}^{\mathrm{T}}\boldsymbol{x} \mid \boldsymbol{A}\boldsymbol{x} \leqslant \boldsymbol{b}, \boldsymbol{l} \leqslant \boldsymbol{x} \leqslant \boldsymbol{u}, x_i \leqslant \lfloor x_i^* \rfloor, x_j \in \mathbb{Z}, \forall j \in \boldsymbol{\mathcal{I}} \right\} \tag{4}$$

$$\underset{\boldsymbol{x}}{\arg\min} \left\{ \boldsymbol{c}^{\mathrm{T}}\boldsymbol{x} \mid \boldsymbol{A}\boldsymbol{x} \leqslant \boldsymbol{b}, \boldsymbol{l} \leqslant \boldsymbol{x} \leqslant \boldsymbol{u}, x_i \geqslant \lceil x_i^* \rceil, x_j \in \mathbb{Z}, \forall j \in \boldsymbol{\mathcal{I}} \right\} \tag{5}$$

where $x_i^*$ represents the value of variable $x_i$ in the optimal solution corresponding to the MILP LP relaxation. Likewise, we branch on the shifted variable $\hat{x}_i$, which generates two sub-problems for AMILP, as represented by the following mathematical expressions:

$$\underset{\hat{\boldsymbol{x}}}{\arg\min} \{ \boldsymbol{c}^{\mathrm{T}}\hat{\boldsymbol{x}} - \boldsymbol{c}^{\mathrm{T}}\boldsymbol{s} \mid \boldsymbol{A}\hat{\boldsymbol{x}} \leq \hat{\boldsymbol{b}}, \hat{\boldsymbol{l}} \leq \hat{\boldsymbol{x}} \leq \hat{\boldsymbol{u}}, \hat{x}_i \leq \lfloor \hat{x}_i^* \rfloor, \hat{x}_j \in \mathbb{Z}, \forall j \in \mathcal{I} \} \tag{6}$$

$$\underset{\hat{\boldsymbol{x}}}{\arg\min} \{ \boldsymbol{c}^{\mathrm{T}}\hat{\boldsymbol{x}} - \boldsymbol{c}^{\mathrm{T}}\boldsymbol{s} \mid \boldsymbol{A}\hat{\boldsymbol{x}} \leq \hat{\boldsymbol{b}}, \hat{\boldsymbol{l}} \leq \hat{\boldsymbol{x}} \leq \hat{\boldsymbol{u}}, \hat{x}_i \geq \lceil \hat{x}_i^* \rceil, \hat{x}_j \in \mathbb{Z}, \forall j \in \mathcal{I} \} \tag{7}$$

where $\hat{x_i}^*$ represents the value of variable $\hat{x}_i$ in the optimal solution corresponding to the AMILP LP relaxation.

According to Lemma 3.1, the LP relaxations of Eq.(4) and Eq.(6) have optimal solutions that can be obtained through variable shifting and these two LP relaxations have equivalent optimal values. Similarly, the optimal values of the LP relaxations of Eq.(5) and Eq.(7) are also equal. Thus, for $\hat{x}_i$ and $x_i$, the lower bound improvements of the subproblems generated from MILP Eq.(1) and AMILP Eq.(3) are equivalent, demonstrating identical branching decisions. The proof is completed. $\square$

Based on Theorem 3.1, it is evident that the generated AMILPs are equipped with expert decision labels, making them readily suitable for imitation learning.

## 3.2 Augmented Bipartite Graph

After obtaining the AMILP, the subsequent task involves constructing the augmented bipartite graph using a modeling approach akin to the one introduced by Gasse et al. (2019). To achieve this, we leverage the above relationship between MILP and AMILP to derive the node features for the augmented bipartite graph from the corresponding node features of the original bipartite graph, as outlined in Table 1. For a detailed overview of the relationships between node features in the augmented and original bipartite graphs, please refer to the Appendix.

### 3.2.1 Constraint Node Features

It is worth noting that the AMILP is derived from a translation transformation of the MILP, resulting in certain invariant features: (1) cosine similarity between constraint coefficients and objective function coefficients; (2) tightness state of the LP relaxation solution within constraints; (3) LP age. Additionally, for the *bias* feature, representing the right-hand term, the transformed feature is $b_i + \boldsymbol{a}_i^{\mathrm{T}} \boldsymbol{s}$. To obtain this term, consider the $i$-th constraint node, which corresponds to the $i$-th constraint of $\boldsymbol{a}_i^{\mathrm{T}} \boldsymbol{x} \leqslant b_i$, After translation, this constraint can be represented as $\boldsymbol{a}_i^{\mathrm{T}} \hat{\boldsymbol{x}} \leqslant b_i + \boldsymbol{a}_i^{\mathrm{T}} \boldsymbol{s}$, leading to the *bias* feature $b_i + \boldsymbol{a}_i^{\mathrm{T}} \boldsymbol{s}$. For *dualsol_val* feature, we consider proposing the following theorem for the explanation.

**Theorem 3.2.** *For MILPs Eq.(1) and AMILPs Eq.(3), let the optimal solution of the dual problem of the LP relaxations be denoted as $\boldsymbol{y}^*$ and $\hat{\boldsymbol{y}}^*$, respectively. Then, a direct correspondence exists between these solutions, indicating that $\boldsymbol{y}^* = \hat{\boldsymbol{y}}^*$.*

The proof can be found in the Appendix. From Theorem 3.2, we can conclude that the *dualsol_val* feature of the augmented bipartite graph remains unchanged compared to the original bipartite graph. Thus, we have successfully determined the constraint node features for the AMILP's bipartite graph through the preceding analysis.

### 3.2.2 Edge Features

Given that an AMILP is generated from the original MIP through variable shifting, the coefficients of the constraints remain invariant throughout this transformation. Consequently, the values of the edge features in the bipartite graph, which directly reflect the coefficients connecting variable nodes and constraint nodes, also remain unchanged.

### 3.2.3 Variable Node Features

Similarly to constraint node features, several variable node features also remain unaltered during the transformation. These include (1) the variable type (i.e., integer or continuous); (2) the coefficients of variables corresponding to the objective function; (3) whether the variable has upper and lower bounds; (4) whether the solution value of a variable is within the bounds; (5) whether the solution value of a variable has a decimal part; (6) the status of the corresponding basic vector; (7) the LP age. For *reduced_cost* features, we consider the following theorem for clarification.

**Theorem 3.3.** *For MILPs Eq.(1) and their corresponding AMILPs Eq.(3), consider the reduced cost corresponding to LP relaxations for a MILP, denoted as $\sigma_i$, and for an AMILP, denoted as $\hat{\sigma}_i$. Then, a direct correspondence exists between these reduced costs, implying that $\sigma_i = \hat{\sigma}_i$.*

The proof is provided in the Appendix. Moreover, the features *sol_val*, *inc_val*, and *avg_inc_val* all exhibit shifts in their values corresponding to the shift vector $s$. With all the above, we have successfully acquired all the features of the augmented bipartite graph.

## 3.3 CONTRASTIVE LEARNING

Contrastive learning has been widely adopted in various domains (He et al., 2020; Chen et al., 2020a; P. et al., 2020; Xu et al., 2022; Iter et al., 2020; Giorgi et al., 2021; Yu et al., 2022). The fundamental idea behind contrastive learning is to pull similar data points (positives) closer together in the feature space while pushing dissimilar ones (negatives) apart. Within our proposed CAMBranch, we leverage this principle by viewing a MILP and its corresponding AMILP as positive pairs while considering the MILP and other AMILPs within the same batch as negative pairs. This enables us to harness the power of contrastive learning to enhance our training process.

We initiate the process with a MILP bipartite graph $(\mathcal{G}_{\text{ori}}, \mathbf{C}_{\text{ori}}, \mathbf{E}_{\text{ori}}, \mathbf{V}_{\text{ori}})$ and its augmented counterpart $(\mathcal{G}_{\text{aug}}, \mathbf{C}_{\text{aug}}, \mathbf{E}_{\text{aug}}, \mathbf{V}_{\text{aug}})$. These graphs undergo processing with a GCNN, following the message passing illustrated in Eq.(2). This results in updated constraint and variable node features $\mathbf{C}'_{\text{ori}}$ and $\mathbf{V}'_{\text{ori}}$ for the MILP, along with $\mathbf{C}'_{\text{aug}}$ and $\mathbf{V}'_{\text{aug}}$ for the AMILP. Subsequently, we generate graph-level representations for both bipartite graphs. To achieve this, we conduct max and average pooling on the constraint nodes and variable nodes, respectively. Merging these embeddings using an MLP yields pooled embeddings for constraint and variable nodes, denoted as $\boldsymbol{c}_{\text{ori}}^{\mathcal{G}}, \boldsymbol{v}_{\text{ori}}^{\mathcal{G}}$ for the MILP and $\boldsymbol{c}_{\text{ori}}^{\mathcal{G}}, \boldsymbol{v}_{\text{ori}}^{\mathcal{G}}$ for the AMILP. These embeddings serve as inputs to another MLP, resulting in graph-level embeddings $\boldsymbol{g}_{\text{ori}}$ and $\boldsymbol{g}_{\text{aug}}$ for MILP and AMILP bipartite graphs, respectively. To train our model using contrastive learning, we treat $\boldsymbol{g}_{\text{ori}}$ and its corresponding $\boldsymbol{g}_{\text{aug}}$ as positive pairs, while considering other AMILPs in the same batch as negative samples. By applying infoNCE (van den Oord et al., 2018) loss, we have

$$\mathcal{L}^{(\text{infoNCE})} = -\sum_{i=1}^{n_{\text{batch}}} \log \left( \frac{\exp\left(\tilde{\boldsymbol{g}}_{\text{ori}}^{\mathrm{T}}(i) \cdot \tilde{\boldsymbol{g}}_{\text{aug}}(i)\right)}{\sum_{j=1}^{n_{\text{batch}}} \exp\left(\tilde{\boldsymbol{g}}_{\text{ori}}^{\mathrm{T}}(i) \cdot \tilde{\boldsymbol{g}}_{\text{aug}}(j)\right)} \right) \tag{8}$$

where $n_{\text{batch}}$ represents the number of samples in a training batch. $\tilde{\boldsymbol{g}}_{\text{ori}}$ and $\tilde{\boldsymbol{g}}_{\text{aug}}$ denote the normalized vectors of $\boldsymbol{g}_{\text{ori}}$ and $\boldsymbol{g}_{\text{aug}}$, respectively. This contrastive learning approach enhances our model's ability to capture representations for MILPs, which further benefits the imitation learning process. The imitation learning process follows Gasse et al. (2019). More details can be found in the Appendix C.3.

## 4 EXPERIMENT

We evaluate our proposed CAMBranch by fully following the settings in Gasse et al. (2019). Due to the space limit, we briefly introduce the experimental setup and results. More details are provided in the Appendix D.

### 4.1 SETUP

**Benchmarks.** Following Gasse et al. (2019), we assess our method on four NP-hard problems, i.e., Set Covering (BALAS, 1980), Combinatorial Auction (Leyton-Brown et al., 2000), Capacitated Facility Location (Cornuejols et al., 1991), and Maximum Independent Set (Cire & Augusto, 2015). Each problem has three levels of difficulty, that is, *Easy*, *Medium*, and *Hard*. We train and test models on each benchmark separately. Through the experiments, we leverage SCIP 6.0.1 (Gleixner et al., 2018) as the backend solver and set the time limit as 1 hour. See more details in the supplementary materials.

**Baselines.** We compare CAMBranch with the following branching strategies: (1) Reliability Pseudocost Branching (RPB) (Achterberg et al., 2005), a state-of-the-art human-designed branching policy and the default branching rule of the SCIP solver; (2) GCNN (Gasse et al., 2019), a state-of-the-art neural branching policy; (3) GCNN (10%), which uses only 10% of the training data from Gasse et al. (2019). This is done to ensure a complete comparison since CAMBranch also utilizes 10% of the data.

**Data Collection and Split.** The expert samples for imitation learning are collected from SCIP rollout with Strong Branching on the *Easy* level instances. Following Gasse et al. (2019), we train

GCNN with 100k expert samples, while CAMBranch is trained with 10% of these samples. Trained with *Easy* level samples, the models are tested on all three level instances. Each level contains 20 new instances for evaluation using five different seeds, resulting in 100 solving attempts for each difficulty level.

**Metrics.** Following Gasse et al. (2019), our metrics are standard for MILP benchmarking, including solving time, number of nodes in the branch and bound search tree, and number of times each method achieves the best solving time among all methods (number of wins). For the first two metrics, smaller values are indicative of better performance, while for the latter, higher values are preferable.

## 4.2 EXPERIMENTAL RESULTS

To assess the effectiveness of our proposed CAMBranch, we conducted the evaluation from two aspects: imitation learning accuracy and MILP instance-solving performance. The former measures the model's ability to imitate the expert strategy, i.e., the Strong Branching strategy. Meanwhile, MILP instance-solving performance evaluates the quality and efficiency of the policy network's decisions, emphasizing critical metrics such as MILP solving time and the size of the B&B tree (i.e., the number of nodes) generated during the solving process.

### 4.2.1 IMITATION LEARNING ACCURACY

First, we initiated our evaluation by comparing CAMBranch with baseline methods in terms of imitation learning accuracy. Following Gasse et al. (2019), we curated a test set comprising 20k expert samples for each problem. The results are depicted in Figure 1. Notably, CAMBranch, trained with 10% of the full training data, outperforms GCNN (10%), demonstrating that our proposed data augmentation and contrastive learning framework benefit the imitation learning process. Moreover, CAMBranch's performance, while exhibiting a slight lag compared to GCNN (Gasse et al., 2019) trained on the entire dataset, aligns with expectations, considering the substantial difference in the size of the training data. CAMBranch delivers comparable performance across three of the four problems, with the exception being the Set Covering problem.

### 4.2.2 INSTANCE SOLVING EVALUATION

Next, we sought to evaluate our proposed CAMBranch on MILP instance solving focusing on three key metrics: solving time, the number of B&B nodes, and the number of wins, as illustrated by Table 2.

Solving time reflects the efficiency of each model in solving MILP instances. As evident in Table 2, we observed that as problem complexity increases, the solving time also rises substantially, with an obvious gap between *Easy* and *Hard* levels. In the *Easy* level, all strategies exhibit similar solving time, with only a maximum gap of about 5 seconds. However, for the *Medium* and *Hard* levels, differences become more significant. Notably, delving into each strategy, we found that, neural network-based policies consistently outperform the traditional RPB, demonstrating the potential of replacing heuristic methods with machine learning-based approaches. Moreover, CAMBranch exhibits the fastest solving process in most cases, particularly in challenging instances. For example, in the hard-level Capacitated Facility Location problem, CAMBranch achieved a solving time of 470.83 seconds, nearly 200 seconds faster than

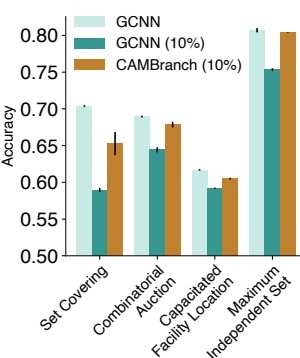

Figure 1: Imitation learning accuracy on the test sets of expert samples.

GCNN. Furthermore, CAMBranch outperforms GCNN (10%) across various instances, reaffirming the effectiveness of our CAMBranch framework.

The number of B&B nodes serves as a measure of branching decision quality, with fewer nodes indicating better decision quality. Table 2 presents similar observations to solving time. CAMBranch consistently outperforms GCNN in most cases, especially in challenging scenarios like the hard-level Maximum Independent Set problem. On average, CAMBranch generates fewer nodes than GCNN (10%), highlighting the efficacy of our data augmentation and contrastive learning network. However, it's worth noting that in some cases, RPB generates the fewest nodes, particularly in

the Capacitated Facility Location problem. Nevertheless, this doesn't translate into shorter solving times, as certain `RPB` decisions are time-consuming.

The number of wins quantifies the instances in which the model achieves the shortest solving time. Higher win counts indicate better performance. With this metric, we examined models at the instance level. From Table 2, we found that GCNN obtains the most times of getting the fastest solving process in the Set Covering problem and the Combinatorial Auction problem (*Easy* and *Medium*). However, for the remaining problems, CAMBranch leads in this metric. Additionally, CAMBranch tends to optimally solve the highest number of instances, except in the case of Set Covering. These results underline the promise of our proposed method, especially in scenarios with limited training data. Collectively, CAMBranch's prominent performance across these three metrics underscores the importance of MILP augmentation and the effectiveness of our contrastive learning framework.

Table 2: Policy evaluation in terms of solving time, number of B&B nodes, and number of wins over number of solved instances on four combinatorial optimization problems. Each level contains 20 instances for evaluation using five different seeds, resulting in 100 solving attempts for each difficulty level. Bold CAMBranch numbers denote column-best results among neural policies.

| Model | Easy | | | Medium | | | Hard | | |
| | Time | Wins | Nodes | Time | Wins | Nodes | Time | Wins | Nodes |
| --- | --- | --- | --- | --- | --- | --- | --- | --- | --- |
| FSB | 17.98 | 0/100 | 27 | 345.91 | 0/90 | 223 | 3600.00 | - | - |
| RPB | 7.82 | 4/100 | 103 | 64.77 | 10/100 | 2587 | 1210.18 | 32 / 63 | 80599 |
| GCNN | 6.03 | 57/100 | 167 | 50.11 | 81/100 | 1999 | 1344.59 | 36/68 | 56252 |
| GCNN (10%) | 6.34 | 39/100 | 230 | 98.20 | 5/96 | 5062 | 2385.23 | 0/6 | 113344 |
| CAMBranch (10%) | 6.79 | 0/100 | 188 | 61.00 | 4/100 | 2339 | 1427.02 | 0/55 | 66943 |
| | | | | Set Covering | | | | | |
| FSB | 4.71 | 0/100 | 10 | 97.6 | 0/100 | 90 | 1396.62 | 0/64 | 381 |
| RPB | 2.61 | 1/100 | 21 | 19.68 | 2/100 | 713 | 142.52 | 29/100 | 8971 |
| GCNN | 1.96 | 43/100 | 87 | 11.30 | 74/100 | 695 | 158.81 | 19/94 | 12089 |
| GCNN (10%) | 1.99 | 44/100 | 102 | 12.38 | 16/100 | 787 | 144.40 | 10/100 | 10031 |
| CAMBranch (10%) | 2.03 | 12/100 | 91 | 12.68 | 8/100 | 758 | **131.79** | **42/100** | **9074** |
| | | | | Combinatorial Auction | | | | | |
| FSB | 34.94 | 0/100 | 54 | 242.51 | 0/100 | 114 | 995.40 | 0/82 | 84 |
| RPB | 30.63 | 9/100 | 79 | 177.25 | 2/100 | 196 | 830.90 | 2/93 | 178 |
| GCNN | 24.72 | 25/100 | 169 | 145.17 | 13/100 | 405 | 680.78 | 5/95 | 449 |
| GCNN (10%) | 26.30 | 15/100 | 180 | 124.49 | 48/100 | 406 | 672.88 | 11/95 | 423 |
| CAMBranch (10%) | 24.91 | **50/100** | 183 | **124.36** | 37/100 | **390** | **470.83** | **77/95** | 428 |
| | | | | Capacitated Facility Location | | | | | |
| FSB | 28.85 | 10/100 | 19 | 1219.15 | 0/62 | 81 | 3600.00 | - | - |
| RPB | 10.73 | 11/100 | 78 | 133.30 | 5/100 | 2917 | 965.67 | 10/40 | 17019 |
| GCNN | 7.17 | 11/100 | 90 | 164.51 | 4/99 | 5041 | 1020.58 | 0/17 | 21925 |
| GCNN (10%) | 7.18 | 26/100 | 103 | 122.65 | 8/89 | 3711 | 695.96 | 2/20 | 17034 |
| CAMBranch (10%) | **6.92** | **42/100** | **90** | **61.51** | **83/100** | 1479 | **496.86** | **33/40** | **10828** |
| | | | | Maximum Independent Set | | | | | |

### 4.2.3 EVALUATION OF DATA COLLECTION EFFICIENCY

In this part, we compared the efficiency of expert sample collection for GCNN in Gasse et al. (2019) and our proposed CAMBranch. We focused on the Capacitated Facility Location Problem, which exhibits the lowest collection efficiency among the four MILP benchmarks and thus closely simulates real-world MILPs by low data collection efficiency. Generating 100k expert samples using Strong Branching to solve the instances takes 84.79 hours. In contrast, if obtaining the same quantity of expert samples, CAMBranch requires 8.48 hours (collecting 10k samples initially) plus 0.28 hours (generating the remaining 90k samples based on the initial 10k), totaling 8.76 hours—an 89.67% time savings. This underscores the superiority of CAMBranch in data collection efficiency.

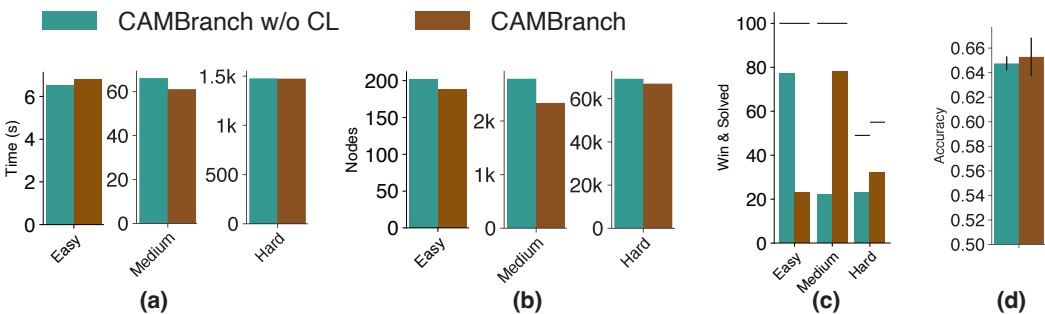

Figure 2: Ablation experiment results on CAMBranch (10%) for instance solving evaluation, including solving time (a), number of nodes (b), and number of wins (c), in addition to imitation learning accuracy (d) for the Set Covering problem.

## 4.3 ABLATION STUDIES

To further validate the effectiveness of contrastive learning, we conducted ablation studies. Specifically, we compared the performance of CAMBranch with contrastive learning to CAMBranch without contrastive learning but with data augmentation, denoted as CAMBranch w/o CL. These experiments were conducted on the Set Cover problem, and the results are displayed in Figure 2. It is evident from the results that integrating contrastive learning significantly enhances CAMBranch's performance, providing compelling evidence of the efficacy of this integration within CAMBranch.

## 4.4 EVALUATING CAMBRANCH ON FULL DATASETS

Previous experiments have showcased CAMBranch's superiority in data-scarce scenarios. To further explore CAMBranch's potential, we conducted evaluations on complete datasets to assess its performance with the entire training data. Table 3 presents the results of instance-solving evaluations for the Combinatorial Auction problem. The outcomes reveal that when trained with the full dataset, CAMBranch (100%) surpasses GCNN (10%), CAMBranch (10%) and even GCNN (100%). Notably, CAMBranch exhibits the fastest solving time for nearly 90% of instances, underscoring its effectiveness. For *Hard* instances, CAMBranch (100%) demonstrates significant improvements across all instance-solving evaluation metrics. These findings affirm that our plug-and-play CAMBranch is versatile, excelling not only in data-limited scenarios but also serving as a valuable tool for data augmentation to enhance performance with complete datasets.

Table 3: The results of evaluating the instance-solving performance for the Combinatorial Auction problem by utilizing the complete training dataset. Bold numbers denote the best results.

| Model | Easy | | | Medium | | | Hard | | |
|---|---|---|---|---|---|---|---|---|---|
| | Time | Wins | Nodes | Time | Wins | Nodes | Time | Wins | Nodes |
| GCNN (10%) | 1.99 | 2/100 | 102 | 12.38 | 3/100 | 787 | 144.40 | 2/100 | 10031 |
| GCNN (100%) | 1.96 | 4/100 | **87** | 11.30 | 7/100 | 695 | 158.81 | 4/94 | 12089 |
| CAMBranch (10%) | 2.03 | 1/100 | 91 | 12.68 | 2/100 | 758 | 131.79 | 11/100 | 9074 |
| CAMBranch (100%) | **1.73** | **93/100** | 88 | **10.04** | **88/100** | 690 | **109.96** | **83/100** | **8260** |

## 5 CONCLUSION

In this paper, we have introduced CAMBranch, a novel framework designed to address the challenge of collecting expert strategy samples for imitation learning when applying machine learning techniques to solve MILPs. By introducing variable shifting, CAMBranch generates AMILPs from the original MILPs, harnessing the collective power of both to enhance imitation learning. Our utilization of contrastive learning enhances the model's capability to capture MILP features, resulting in more effective branching decisions. We have evaluated our method on four representative combinatorial optimization problems and observed that CAMBranch exhibits superior performance, even when trained on only 10% of the complete dataset. This underscores the potential of CAMBranch, especially in scenarios with limited training data.

ACKNOWLEDGEMENT

This work is supported by National Key R&D Program of China (Grant No.2021YFC2902701).

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

**Contents**

# A  BACKGROUND

## A.1  BRANCH AND BOUND (B&B)

The B&B algorithm operates in an iterative way at each node. It commences with solving the LP relaxation of the original MILP problem Eq. (1). This LP relaxation serves as a lower bound estimate for the original MILP (1). If the optimal solution $\boldsymbol{x}^*$ of this LP relaxation satisfies all the constraints in (1), its objective value provides an upper bound estimate for the original problem. However, if $\boldsymbol{x}^*$ contains non-integer variables $x_j^*$ (where $j \in \mathcal{I}$), the algorithm proceeds by splitting the problem into two sub-problems, each with additional constraints: $x_j \leq \lfloor x_j^* \rfloor$ and $x_j \geq \lceil x_j^* \rceil$, where $\lfloor . \rfloor$ and $\lceil . \rceil$ represent the floor and ceiling functions, respectively.

Throughout this iterative process, any sub-problem that exceeds the upper bound estimate is pruned, eliminating the need for further branching. This pruning strategy significantly enhances computational efficiency. The algorithm repeats the above process until either all nodes are pruned or the optimality gap reaches a predetermined threshold. At this point, the global optimal solution, along with its corresponding objective value, is successfully obtained.

## A.2  STRONG BRANCHING

The core idea behind the Strong Branching strategy (Applegate et al., 1995) is to identify the candidate variable that, when chosen for branching, provides the maximum improvement in the lower bound of the problem. Specifically, this strategy involves pre-branching all candidate variables, solving their respective Linear Programming (LP) relaxation problems, and then selecting the variable that contributes the most to lower bound enhancement as the actual branching variable. When pre-branching is applied to all available candidate variables, and each LP relaxation problem is solved to optimality, this strategy is known as Full Strong Branching (FSB), which is the expert strategy for imitation learning in this paper. Essentially, Full Strong Branching can be seen as a greedy approach aimed at identifying the locally optimal variable for branching.

Full Strong Branching often leads to the smallest branching search tree. However, it comes at a considerable computational cost. Consequently, several variants of Full Strong Branching have been developed to mitigate this computational burden. These variants involve selecting only a subset of candidate variables for branching or limiting the number of iterations during the solution of each LP relaxation problem. It's important to note that while these modifications enhance computational efficiency, they do not fundamentally reduce the computational complexity inherent in the Strong Branching strategy.

# B  PROOFS

**Lemma 3.1.** *For MILPs (Eq.(1)) and AMILPs (Eq.(3)), let the LP relaxation optimal solutions be denoted as $\boldsymbol{x}^*$ and $\hat{\boldsymbol{x}}^*$, respectively. Then, there exists a correspondence between these solutions such that $\hat{\boldsymbol{x}}^* = \boldsymbol{x}^* + \boldsymbol{s}$.*

*Proof.* The LP relaxation problems of MILP and AMILP can be expressed as follows:

$$\boldsymbol{x}^* = \arg\min_{\boldsymbol{x}} \left\{ \boldsymbol{c}^{\mathrm{T}} \boldsymbol{x} \mid \boldsymbol{A}\boldsymbol{x} \leqslant \boldsymbol{b}, \boldsymbol{l} \leqslant \boldsymbol{x} \leqslant \boldsymbol{u} \right\} \tag{9}$$

and

$$\hat{\boldsymbol{x}}^* = \arg\min_{\hat{\boldsymbol{x}}} \left\{ \boldsymbol{c}^{\mathrm{T}} \hat{\boldsymbol{x}} - \boldsymbol{c}^{\mathrm{T}} \boldsymbol{s} \mid \boldsymbol{A}\hat{\boldsymbol{x}} \leqslant \hat{\boldsymbol{b}}, \hat{\boldsymbol{l}} \leqslant \hat{\boldsymbol{x}} \leqslant \hat{\boldsymbol{u}} \right\} \tag{10}$$

We now assume that $\boldsymbol{y} = \boldsymbol{x}^* + \boldsymbol{s}$, so that Eq.(9) can be written as

$$\boldsymbol{y} - \boldsymbol{s} = \arg\min_{\boldsymbol{x}} \left\{ \boldsymbol{c}^{\mathrm{T}} \boldsymbol{x} \mid \boldsymbol{A}\boldsymbol{x} \leqslant \boldsymbol{b}, \boldsymbol{l} \leqslant \boldsymbol{x} \leqslant \boldsymbol{u} \right\} \tag{11}$$

which implies that

$$
\begin{aligned}
\boldsymbol{y} &= \arg\min_{\boldsymbol{x}} \left\{ \boldsymbol{c}^{\mathrm{T}}\boldsymbol{x} \mid \boldsymbol{A}\boldsymbol{x} \leqslant \boldsymbol{b}, \boldsymbol{l} \leqslant \boldsymbol{x} \leqslant \boldsymbol{u} \right\} + \boldsymbol{s} \\
&= \arg\min_{\boldsymbol{x}} \left\{ \boldsymbol{c}^{\mathrm{T}}(\hat{\boldsymbol{x}} - \boldsymbol{s}) \mid A(\hat{\boldsymbol{x}} - \boldsymbol{s}) \leqslant b, \boldsymbol{l} \leqslant \hat{\boldsymbol{x}} - \boldsymbol{s} \leq u \right\} + \boldsymbol{s} \\
&= \arg\min_{\boldsymbol{x}+\boldsymbol{s}} \left\{ \boldsymbol{c}^{\mathrm{T}}\hat{\boldsymbol{x}} - \boldsymbol{c}^{\mathrm{T}}\boldsymbol{s} \mid A\hat{\boldsymbol{x}} \leq As + b, \boldsymbol{l} + \boldsymbol{s} \leqslant \hat{\boldsymbol{x}} \leqslant u + \boldsymbol{s} \right\}
\end{aligned}
\tag{12}
$$

Clearly, Eq.(10) is equal to Eq.(12), that is, $\hat{\boldsymbol{x}}^* = \boldsymbol{x}^* + \boldsymbol{s}$. $\qquad\square$

**Theorem 3.1.** *For MILPs Eq.(1) and AMILPs Eq.(3), let the optimal solution of the dual problem of the LP relaxation solution be denoted as $\boldsymbol{y}^*$ and $\hat{\boldsymbol{y}}^*$, respectively. Then, there exists a correspondence between these solutions such that $\boldsymbol{y}^* = \hat{\boldsymbol{y}}^*$.*

*Proof.* By incorporating the boundary constraints of variable $\boldsymbol{x}$ into the set of other constraints, the MILP problem can be reformulated as the following LP relaxation form

$$
\boldsymbol{x}^* = \arg\min_{\boldsymbol{x}} \{ \boldsymbol{c}^{\mathrm{T}}\boldsymbol{x} | \bar{\boldsymbol{A}}\boldsymbol{x} \leq \boldsymbol{b} \}
\tag{13}
$$

We can obtain its dual problem as follows

$$
\boldsymbol{y}^* = \arg\max_{\boldsymbol{y}} \{ -\boldsymbol{y}^{\mathrm{T}}\boldsymbol{b} | \boldsymbol{c}^{\mathrm{T}} + \boldsymbol{y}^{\mathrm{T}}\bar{\boldsymbol{A}} = \boldsymbol{0}, \boldsymbol{y} \geq \boldsymbol{0} \}
\tag{14}
$$

Similarly, for AMILPs, we also combine the boundary constraints of variable $\hat{\boldsymbol{x}}^*$ with other constraints, then we have

$$
\hat{\boldsymbol{x}}^* = \arg\min_{\hat{x}} \{ \boldsymbol{c}^{\mathrm{T}}\hat{\boldsymbol{x}} - \boldsymbol{c}^{\mathrm{T}}\boldsymbol{s} | \bar{\boldsymbol{A}}\hat{\boldsymbol{x}} \leq \boldsymbol{b} + \bar{\boldsymbol{A}}\boldsymbol{s} \}
\tag{15}
$$

The dual problem of this AMILP can be obtained as follows:

$$
\begin{aligned}
\hat{\boldsymbol{y}}^* &= \arg\max_{\hat{\boldsymbol{y}}} \{ -\hat{\boldsymbol{y}}^{\mathrm{T}}(\boldsymbol{b} + \bar{\boldsymbol{A}}\boldsymbol{s}) - \boldsymbol{c}^{\mathrm{T}}\boldsymbol{s} | \boldsymbol{c}^{\mathrm{T}} + \hat{\boldsymbol{y}}^{\mathrm{T}}\bar{\boldsymbol{A}} = \boldsymbol{0}, \hat{\boldsymbol{y}} \geq \boldsymbol{0} \} \\
&= \arg\max_{\hat{\boldsymbol{y}}} \{ -\hat{\boldsymbol{y}}^{\mathrm{T}}\boldsymbol{b} | \boldsymbol{c}^{\mathrm{T}} + \hat{\boldsymbol{y}}^{\mathrm{T}}\bar{\boldsymbol{A}} = \boldsymbol{0}, \hat{\boldsymbol{y}} \geq \boldsymbol{0} \}
\end{aligned}
\tag{16}
$$

According to Eq.(14) and Eq.(16), it can be inferred that the optimal solutions for the dual problem associated with LP relaxation of both MILP and AMILP is identical. The proof is completed. $\qquad\square$

**Theorem 3.3.** *For MILPs Eq.(1) and their corresponding AMILPs Eq.(3), let the reduced cost corresponding to LP relaxations for a MILP and an AMILP be denoted as $\sigma_i$ and $\hat{\sigma}_i$, respectively. Then, there is a correspondence between these reduced costs such that $\sigma_i = \hat{\sigma}_i$.*

*Proof.* Let's consider the variable $x_j$ in the MILP. Its reduced cost, denoted as $\sigma_j$, can be expressed as

$$
\sigma_j = c_j - \boldsymbol{c}_B^{\mathrm{T}}\boldsymbol{B}^{-1}\hat{\boldsymbol{A}}_j
\tag{17}
$$

Here, $\boldsymbol{B}$ represents the matrix composed of the column vectors of the current basis, $\boldsymbol{c}_B$ denotes the coefficient vector of the objective function at basic variables, and $\hat{\boldsymbol{A}}_j$ represents the $j$th column of the matrix $\hat{\boldsymbol{A}}$. Similarly, for the AMILP, the reduced cost of variable $\hat{\boldsymbol{x}}$, denoted as $\hat{\sigma}_j$, can be expressed in the same way:

$$
\hat{\sigma}_j = c_j - \boldsymbol{c}_B^{\mathrm{T}}\boldsymbol{B}^{-1}\hat{\boldsymbol{A}}_j = \sigma_j
\tag{18}
$$

This equality demonstrates that the reduced cost values for the variables in both the MILP and the AMILP are indeed equivalent. Hence, the proof is complete. $\qquad\square$

## C  METHODOLOGY DETAILS

### C.1  OVERVIEW OF BIPARTITE GRAPH NODE FEATURES

The relationships between MILP and AMILP node features are illustrated in Table 4 for constraint nodes and Table 5 for variable nodes. Note that in Table 5, the notation $\mathbb{B} \leftrightarrow \mathbb{Z}$ denotes the potential mutual conversion between binary variables and integer variables.

Table 4: Relationship between MILP and AMILP constraint node features.

| Node feature of constraint $i$ | MILP | AMILP |
|---|---|---|
| obj_cos_sim | $\mathbf{C}_{i,1}$ | $\mathbf{C}_{i,1}$ |
| bias | $\mathbf{C}_{i,2}$ | $\mathbf{C}_{i,2} + \boldsymbol{a}_i^{\mathrm{T}} \boldsymbol{s}$ |
| is_tight | $\mathbf{C}_{i,3}$ | $\mathbf{C}_{i,3}$ |
| dualsol_val | $\mathbf{C}_{i,4}$ | $\mathbf{C}_{i,4}$ |
| age | $\mathbf{C}_{i,5}$ | $\mathbf{C}_{i,5}$ |

Table 5: Relationship between MILP and AMILP variable node features.

| Node feature of variable $j$ | MILP | AMILP |
|---|---|---|
| type | $\mathbf{V}_{j,1}$ | $\mathbf{V}_{j,1}$ or $\mathbb{B} \leftrightarrow \mathbb{Z}$ |
| coef | $\mathbf{V}_{j,2}$ | $\mathbf{V}_{j,2}$ |
| has_lb | $\mathbf{V}_{j,3}$ | $\mathbf{V}_{j,3}$ |
| has_ub | $\mathbf{V}_{j,4}$ | $\mathbf{V}_{j,4}$ |
| sol_is_at_lb | $\mathbf{V}_{j,5}$ | $\mathbf{V}_{j,5}$ |
| sol_is_at_ub | $\mathbf{V}_{j,6}$ | $\mathbf{V}_{j,6}$ |
| sol_frac | $\mathbf{V}_{j,7}$ | $\mathbf{V}_{j,7}$ |
| basis_status | $\mathbf{V}_{j,8}$ | $\mathbf{V}_{j,8}$ |
| reduced_cost | $\mathbf{V}_{j,9}$ | $\mathbf{V}_{j,9}$ |
| age | $\mathbf{V}_{j,10}$ | $\mathbf{V}_{j,10}$ |
| sol_val | $\mathbf{V}_{j,11}$ | $\mathbf{V}_{j,11} + s_j$ |
| inc_val | $\mathbf{V}_{j,12}$ | $\mathbf{V}_{j,11} + s_j$ |
| avg_inc_val | $\mathbf{V}_{j,13}$ | $\mathbf{V}_{j,13} + s_j$ |

## C.2 CONTRASTIVE LEARNING

This section will provide details the forward propagation process. Specifically, once we have acquired the updated constraint and variable node features $\mathbf{C}'_{\mathrm{ori}}$ and $\mathbf{V}'_{\mathrm{ori}}$ for the MILP, along with $\mathbf{C}'_{\mathrm{aug}}$ and $\mathbf{V}'_{\mathrm{aug}}$ for the AMILP, we can formulate the feature merging process as follows:

$$\boldsymbol{c}^{\mathcal{G}}_{\mathrm{ori}} = \mathrm{MLP}\left(\mathrm{Concat}\left(\mathrm{MaxPool}\left(\mathbf{C}'_{\mathrm{ori}}\right), \mathrm{MeanPool}\left(\mathbf{C}'_{\mathrm{ori}}\right)\right)\right) \tag{19}$$

$$\boldsymbol{v}^{\mathcal{G}}_{\mathrm{ori}} = \mathrm{MLP}\left(\mathrm{Concat}\left(\mathrm{MaxPool}\left(\mathbf{V}'_{\mathrm{ori}}\right), \mathrm{MeanPool}\left(\mathbf{V}'_{\mathrm{ori}}\right)\right)\right) \tag{20}$$

$$\boldsymbol{c}^{\mathcal{G}}_{\mathrm{aug}} = \mathrm{MLP}\left(\mathrm{Concat}\left(\mathrm{MaxPool}\left(\mathbf{C}'_{\mathrm{aug}}\right), \mathrm{MeanPool}\left(\mathbf{C}'_{\mathrm{aug}}\right)\right)\right) \tag{21}$$

$$\boldsymbol{v}^{\mathcal{G}}_{\mathrm{aug}} = \mathrm{MLP}\left(\mathrm{Concat}\left(\mathrm{MaxPool}\left(\mathbf{V}'_{\mathrm{aug}}\right), \mathrm{MeanPool}\left(\mathbf{V}'_{\mathrm{aug}}\right)\right)\right) \tag{22}$$

where Concat denotes the concatenation operation. The process of obtaining graph-level embeddings can be formulated as follows:

$$\boldsymbol{g}_{\mathrm{ori}} = \mathrm{MLP}\left(\mathrm{Concat}\left(\boldsymbol{c}^{\mathcal{G}}_{\mathrm{ori}}, \boldsymbol{v}^{\mathcal{G}}_{\mathrm{ori}}\right)\right) \quad \boldsymbol{g}_{\mathrm{aug}} = \mathrm{MLP}\left(\mathrm{Concat}\left(\boldsymbol{c}^{\mathcal{G}}_{\mathrm{aug}}, \boldsymbol{v}^{\mathcal{G}}_{\mathrm{aug}}\right)\right) \tag{23}$$

where $\boldsymbol{g}_{\mathrm{ori}}$ and $\boldsymbol{g}_{\mathrm{aug}}$ represent the graph-level embeddings of the MILP bipartite graph and the AMILP bipartite graph, respectively. Once these graph-level embeddings are obtained, we proceed to apply contrastive learning in the subsequent steps.

### C.3 Imitation Learning Training Procedure

We employ behavior cloning Pomerleau (1991) to train the CAMBranch, focusing on imitating Strong Branching policies. Expert strategies are collected using the optimization suite SCIP Gleixner et al. (2018) and are stored in a dataset that consists of expert state-action pairs, denoted as $\mathcal{D} = (\mathbf{s}_i, \mathbf{a}_i^*)_{i=1}^N$. Within this dataset, $\mathbf{s}_i = (\mathcal{G}, \mathbf{C}, \mathbf{E}, \mathbf{V})$, while $\mathbf{a}_i^*$ represents the branch decision of Strong Branching strategy under $s_t$. To optimize the network, cross-entropy is used as a supervised learning loss function

$$\mathcal{L}^{(\text{sup})} = -\frac{1}{N} \sum_{(\mathbf{s}_i, \mathbf{a}_i^*) \in D} \log \pi_\theta (\mathbf{a}_i^* \mid \mathbf{s}_i) \tag{24}$$

In addition, to enhance the consistency in the probability distribution of variable selection between the MILP and the AMILP, we incorporate consistency constraints. Specifically, we extract the probability distribution $\boldsymbol{P}_{\text{ori}}$ and $\boldsymbol{P}_{\text{aug}}$ of variables outputted by MILP and the AMILP, respectively. The aim is to minimize the divergence between these two distributions, i.e.

$$\mathcal{L}^{(\text{Aux})} = \sum_{i=1}^{n_{\text{batch}}} (\boldsymbol{P}_{\text{ori}}(i) - \boldsymbol{P}_{\text{aug}}(i))^2 \tag{25}$$

Finally, the final loss function by combining the three loss above is obtained

$$\mathcal{L} = \mathcal{L}^{(\text{sup})} + \lambda_1 \mathcal{L}^{(\text{infoNCE})} + \lambda_2 \mathcal{L}^{(\text{Aux})} \tag{26}$$

Where $\lambda_1$ and $\lambda_2$ are hyperparameters used to adjust the weights between loss functions.

## D Experiment Details

### D.1 Experimental Settings

In this paper, all experiments are run on a cluster with Intel(R) Xeon(R) Gold 5218 CPU @ 2.30GHz processors, 128GB RAM, and Nvidia RTX 2080Ti graphics cards.

#### D.1.1 Data Collection and Split

In our experiments, we conducted a systematic investigation of our model's performance by categorizing each problem instance into three difficulty levels: *Easy*, *Medium*, and *Hard*, following the established settings used in Gasse et al. (2019). Additional information about the instance data split for each problem can be found in Gasse et al. (2019).

#### D.1.2 Solver Configuration

In our experiments, we employed the open-source solver SCIP (version 6.0.1) (Gleixner et al., 2018) as our backend solver. We imposed a maximum solving time limit of 3600 seconds, allowing cut generation operations solely at the root node while disabling solver restarts. To ensure a fair comparison among the methods, we maintained all other solver parameters at their default values Khalil et al. (2016); Kilinç-Karzan et al. (2009); Matteo Fischetti (2012).

#### D.1.3 Training Parameters

In our experiments, we implemented models using PyTorch (Paszke et al., 2019) and PyTorch Geometric (Fey & Lenssen, 2019). We utilized the Adam optimizer Kingma & Ba (2015) with $\beta_1 = 0.9$ and $\beta_2 = 0.999$. In case the model shows no significant improvement over a period of 10 epochs, we applied a learning rate reduction to 20% of its initial value. We set the hidden layer size of the GCNN network to 64. We conducted a grid search for the learning rate, considering values from $\{1 \times 10^{-3}, 5 \times 10^{-4}, 1 \times 10^{-4}\}$. Additionally, we selected the weight values $\lambda_1 = 0.05$ and $\lambda_2 = 0.01$ for the loss function.

### D.1.4 METRICS

We evaluate performance using the shift geometric mean (SGM) for solving time and the geometric mean of nodes for the number of B&B search tree nodes. The SGM is calculated for a set of $n$ numbers, denoted as $t_1, t_2, \ldots, t_n$, with $s$ representing the shift. The SGM formula is given by $\mathrm{SGM} = \sqrt[n]{\prod_{i=1}^{n}(t_i + s)} - s$. In this context, smaller values for the solving time and node count metrics indicate better performance, while a higher number of wins metric signifies superior performance. In this paper, $s$ is set to 1 for time and 100 for Nodes, following the previous work (Gasse et al., 2019; Zarpellon et al., 2021).

### D.2 ADDITIONAL RESULTS

Our results are also presented in a tabular format, available in Table 6 for imitation learning accuracy, Table 7 and Table 8 for the ablation study. Additionally, we delve into the influence of varying the training sample ratio on performance. Given that our proposed CAMBranch requires additional computations compared to the original GCNN in Gasse et al. (2019), we also evaluated the training overhead of these models.

Table 6: Imitation learning accuracy on the test sets of expert samples.

| Model | Set Covering | | | Combinatorial Auction | | | Capacitated Facility Location | | | Maximum Independent Set | | |
|---|---|---|---|---|---|---|---|---|---|---|---|---|
| | acc@1 | acc@5 | acc@10 | acc@1 | acc@5 | acc@10 | acc@1 | acc@5 | acc@10 | acc@1 | acc@5 | acc@10 |
| GCNN | 70.39 ± 0.28 | 93.09 ± 0.14 | 98.40 ± 0.09 | 68.95 ± 0.32 | 92.79 ± 0.16 | 97.87 ± 0.09 | 61.70 ± 0.23 | 95.48 ± 0.12 | 99.68 ± 0.01 | 80.70 ± 0.72 | 92.78 ± 0.19 | 95.83 ± 0.22 |
| GCNN (10 %) | 58.98 ± 0.69 | 82.97 ± 0.52 | 91.61 ± 0.44 | 64.44 ± 0.90 | 90.37 ± 0.17 | 96.70 ± 0.08 | 59.20 ± 0.21 | 95.10 ± 0.08 | 99.68 ± 0.02 | 75.36 ± 0.45 | 90.84 ± 0.39 | 94.39 ± 0.30 |
| CAMBranch | 65.27 ± 3.92 | 89.46 ± 2.98 | 96.14 ± 1.90 | 67.85 ± 0.97 | 91.52 ± 0.50 | 97.14 ± 0.39 | 60.50 ± 0.30 | 95.24 ± 0.13 | 99.64 ± 0.01 | 80.38 ± 0.11 | 92.34 ± 0.20 | 95.47 ± 0.20 |

Table 7: Results of ablation experiments on imitation learning accuracy in Set Covering.

| Type | acc @1 | acc @5 | acc @10 |
|---|---|---|---|
| CAMBranch w/o CL (10%) | 64.75 ± 1.43 | 88.94 ± 1.31 | 96.16 ± 0.72 |
| CAMBranch (10%) | 65.27 ± 3.92 | 89.46 ± 2.98 | 96.14 ± 1.90 |

### D.2.1 ANALYSIS OF IMPACT OF TRAINING SAMPLE RATIOS

To investigate the influence of training sample ratios, we conducted experiments using subsets comprising 5%, 10%, and 20% of the training data for both GCNN and CAMBranch. We evaluated their performance on the Combinatorial Auction problem, and the results are summarized in Table 9. Our observations indicate that as the problem's difficulty level increases, the performance differences among the training sample ratios become progressively obvious. Moreover, for both GCNN and CAMBranch, their best performances are consistently achieved when the training sample ratio is set at 20%, suggesting that leveraging a larger portion of data leads to performance improvement. Similar trends are evident on the Maximum Independent Set problem (Table 10), especially in the *Hard* level instances. As shown in Table 10, the performance on *Hard* instances improves with larger sample sizes, with reduced solving time and a lower number of nodes. Notably, CAMBranch (20%) achieved the highest number of solved instances in the *Hard* level, 49 instances in total, underscoring its superiority. Furthermore, the comparison between the two methods reveals that CAMBranch consistently outperforms GCNN across all three ratios, highlighting the superior capabilities of our proposed CAMBranch framework.

Table 8: Results of ablation experiments on instance solving evaluation in Set Covering.

| Model | Easy | | | Medium | | | Hard | | |
|---|---|---|---|---|---|---|---|---|---|
| | Time | Wins | Nodes | Time | Wins | Nodes | Time | Wins | Nodes |
| CAMBranch w/o CL (10%) | 6.55 | 77/100 | 202 | 66.09 | 22/100 | 2788 | 1472.84 | 23/49 | 69215 |
| CAMBranch (10%) | 6.79 | 23/100 | 188 | 61.00 | 78/100 | 2339 | 1427.02 | 32/55 | 66943 |

Table 9: The results of sample ratio analysis on the instance solving evaluation for Combinatorial Auction.

| Model | Easy | | | Medium | | | Hard | | |
|---|---|---|---|---|---|---|---|---|---|
| | Time | Wins | Nodes | Time | Wins | Nodes | Time | Wins | Nodes |
| GCNN (5%) | 1.61 | 37/100 | 95 | 10.66 | 7/100 | 803 | 117.90 | 5/100 | 10692 |
| GCNN (10%) | 1.99 | 1/100 | 102 | 12.38 | 0/100 | 787 | 144.40 | 0/100 | 10031 |
| GCNN (20%) | 1.66 | 15/100 | 94 | 9.81 | 51/100 | 796 | 107.02 | 27/100 | 9626 |
| CAMBranch (5%) | 1.61 | 38/100 | 97 | 10.64 | 12/100 | 825 | 121.81 | 2/100 | 10582 |
| CAMBranch (10%) | 2.03 | 0/100 | 91 | 12.68 | 0/100 | 758 | 131.79 | 2/100 | 9074 |
| CAMBranch (20%) | 1.68 | 9/100 | 91 | 9.92 | 30/100 | 762 | 103.38 | 64/100 | 9050 |

Table 10: The results of sample ratio analysis on the instance solving evaluation for Maximum Independent Set.

| Model | Easy | | | Medium | | | Hard | | |
|---|---|---|---|---|---|---|---|---|---|
| | Time | Wins | Nodes | Time | Wins | Nodes | Time | Wins | Nodes |
| GCNN (5%) | 5.97 | 9/100 | 90 | 64.07 | 13/100 | 1824 | 607.48 | 2/39 | 16850 |
| GCNN (10%) | 7.18 | 0/100 | 103 | 51.40 | 37/100 | 1331 | 695.96 | 0/20 | 17034 |
| GCNN (20%) | 5.84 | 22/100 | 88 | 55.22 | 26/98 | 1534 | 465.00 | 16/43 | 12998 |
| CAMBranch (5%) | 5.85 | 65/100 | 92 | 70.91 | 3/100 | 1982 | 592.81 | 1/41 | 15480 |
| CAMBranch (10%) | 6.92 | 0/100 | 90 | 61.51 | 14/100 | 1479 | 496.86 | 1/40 | 10828 |
| CAMBranch (20%) | 6.19 | 4/100 | 95 | 68.47 | 7/100 | 2008 | 416.24 | 20/49 | 10455 |

### D.2.2 ANALYSIS OF TRAINING OVERHEAD

Since our proposed CAMBranch introduces additional computation, we further assessed the training overhead of CAMBranch and GCNN proposed in Gasse et al. (2019). Table 11 demonstrates that, despite the introduced computation by CAMBranch, the observed difference in computational speed is deemed acceptable—merely a matter of several milliseconds. In summary, the training overhead associated with CAMBranch is within acceptable bounds.

## E DISCUSSION

### E.1 WHEN TO USE CAMBRANCH?

In this section, we further explore the scenarios in which CAMBranch is most effective. To this end, we conducted the following experiments on Combinatorial Auction Problem to analyze the relationships between performance and both the number of MILP instances and expert samples. The results of these experiments are detailed in Table 12.

**Impact of expert sample size.** From experiment pairs (I, II), (III, IV), and (V, VI), we delve into the relationship between performance and the number of expert samples, holding the number of MILP instances constant. The results indicate that increasing the number of expert samples generally enhances performance, as evident in pairs (I, II) and (III, IV). However, in scenarios with a relatively small number of instances, such as (V, VI), augmenting expert sample size may not necessarily lead to a performance increase.

**Influence of MILP instance count.** Examining pairs (I, III, IV) and (II, IV, VI), we further explore the relationship between performance and the number of MILP instances, maintaining a consistent expert sample size. The observations suggest a general trend of performance improvement with an increase in the number of MILP instances.

Based on these observations, to answer the question of when to use CAMBranch, our initial conclusion is that CAMBranch demonstrates more potential when there is a sufficient number of MILP instances, but not necessarily too large scale. Using several hundreds of MILP instances can already

Table 11: Results of training time per batch of GCNN and CAMBranch on the Set Covering problem. Each batch contains 64 samples.

| Model | Training time per batch (s) |
|---|---|
| CAMBranch (10%) | 0.09666 |
| GCNN (10%) | 0.09216 |

Table 12: Results for the impacts of expert sample size and MILP instance count.

| Exp ID | Training Data | Easy | | Medium | | Hard | |
|---|---|---|---|---|---|---|---|
| | | Time | Nodes | Time | Nodes | Time | Nodes |
| I | 362 instances 10k expert samples | 2.03 | 91 | 12.68 | 758 | 131.79 | 9074 |
| II | 388 instances 20k expert samples | 1.68 | 91 | 9.92 | 762 | 103.38 | 9050 |
| III | 100 instances 10k expert samples | 1.67 | 94 | 10.02 | 757 | 111.52 | 9242 |
| IV | 100 instances 20k expert samples | 1.64 | 90 | 10.03 | 751 | 109.58 | 9195 |
| V | 50 instances 10k expert samples | 3.25 | 98 | 20.58 | 796 | 158.24 | 9834 |
| VI | 50 instances 20k expert samples | 3.18 | 97 | 20.59 | 819 | 166.02 | 10892 |

achieve superior performance. Moreover, increasing the number of expert samples provides benefits for performance, as supported by our empirical findings.

## E.2    RELATIONSHIP BETWEEN IMITATION LEARNING AND INSTANCE EVALUATION

In Section 4.2, we presented the results of various branching strategies. However, due to space limitations, we aim to delve deeper into these results in this discussion section. Specifically, we'll focus on the relationship between the results of imitation learning accuracy and instance evaluation.

Regarding imitation learning accuracy, it's notable that CAMBranch doesn't consistently outperform GCNN in imitating the Strong Branching strategy. However, when we shift our attention to instance evaluation, CAMBranch exhibits superior performance in most cases, particularly on challenging problems, with the exception of Set Covering. This intriguing observation prompts us to explore the underlying reasons. Analyzing CAMBranch's training signal, i.e., loss function, one possible explanation is that contrastive learning empowers CAMBranch to transcend mere imitation of Strong Branching. While Strong Branching often results in the creation of the smallest B&B trees due to its tendency to produce high-quality branching decisions, it doesn't guarantee an optimal branching strategy. Recent studies by Scavuzzo et al. (2022), Dey et al. (2023), and Gamrath et al. (2020) have even suggested that Strong Branching may underperform in certain cases, falling short of problem-specific rules.

Within our CAMBranch framework, the introduction of contrastive learning offers a key advantage. It enables CAMBranch to learn MILP representations better, allowing it to uncover alternative branching strategies beyond Strong Branching. This means that CAMBranch can now explore and learn from alternative policies that might yield better performance than Strong Branching, guided by the contrastive learning loss function. Essentially, our proposed framework serves as a pathway to learning potentially superior branching strategies, extending beyond the confines of merely imitating Strong Branching. Therefore, this will enhance decision-making capabilities in the realm of MILP solving. We will delve deeper into this aspect in our future research.

## F    RELATED WORK

**Learning to branch.**    The integration of machine learning techniques into branching strategies has been a topic of growing interest, with Gasse et al. (2019) marking a significant breakthrough in this domain, signifying a pivotal moment. Prior to this, conventional approaches in relevant literature (Khalil et al., 2016; Alvarez et al., 2017; Balcan et al., 2018) often rely on extracting statistical features from MILPs during the solving process. However, these feature extraction methods were deemed incomplete for capturing the full essence of MILP problem instances. By contrast, the work of Gasse et al. (2019) introduced a novel perspective by leveraging bipartite graphs to create more accurate modeling of MILPs and trained models with imitation learning to approximate the Strong Branching. Subsequent research efforts predominantly build upon the foundations laid by Gasse et al. (2019). For instance, Gupta et al. (2020) proposed a hybrid model tailored for efficient branching on CPU-constrained machines. This model replaced computationally expensive graph networks with more lightweight Multilayer Perceptrons (MLPs), except at the root node. Meanwhile, Nair et al. (2020) achieved notable enhancements in both runtime performance and the average primal-dual gap by introducing Neural Diving and Neural Branching techniques.

While these methods demonstrated impressive results, they were primarily designed for homogeneous MILPs, where training and test instances belong to the same problem class. In order to extend the applicability to heterogeneous MILPs, where problem instances vary across different classes, Zarpellon et al. (2021) and Lin et al. (2022) adopted approaches that parameterize the states of B&B search trees. They achieved this by imitating the SCIP default branching scheme known as RPB, a suitable expert policy renowned for its effectiveness in guiding search trees. For more comprehensive insights into this field, readers can refer to related surveys by Khalil et al. (2017) and Zhang et al. (2023).

**Contrastive learning.**    Contrastive learning has emerged as a powerful paradigm in various machine learning domains. It has been successfully applied in computer vision (Chen et al., 2020a; Falcon & Cho, 2020; He et al., 2020; Chen et al., 2020c;b; Caron et al., 2020; Grill et al., 2020; Chen & He, 2021), natural language processing (Mikolov et al., 2013; Arora et al., 2019; Iter et al., 2020; Fang & Xie, 2020; Giorgi et al., 2021), recommendation system (Yu et al., 2022) and drug discovery (Xu et al., 2022). The core idea involves encouraging similarity between positive pairs (similar data points) and pushing apart negative pairs (dissimilar data points) in a latent space. In this paper, CAMBranch leverages the contrastive learning paradigm to maximize the utilization of both MILPs and their corresponding AMILP data.

## G    MILP FORMULATION

Following Gasse et al. (2019), we evaluated the models on four combinatorial optimization problems, i.e., Set Covering, Combinatorial Auction, Capacitated Facility Location, and Maximum Independent Set represent. These problem classes have served as standard benchmarks for evaluating the efficacy of optimization techniques in the research community. Despite considerable attention and research efforts, these problems remain computationally demanding, even for state-of-the-art solvers. In the following sections, we provide detailed descriptions of the MILP models for each of these problems.

(1) Set Covering: given a finite set $\mathcal{U}$ and its $n$ subsets $S_1, \cdots, S_n$, the problem seeks to identify the minimum number of subsets that can be used to cover $\mathcal{U}$ completely.

$$\text{minimize} \quad \sum_{j=1}^{n} x_j \tag{27}$$

$$\text{subject to} \quad \sum_{j \in \{1, \cdots, n\} \mid v \in S_j} x_j \geqslant 1 \quad \forall v \in \mathcal{U} \tag{28}$$

$$x_j \in \{0, 1\} \quad \forall j \in \{1, \cdots, n\} \tag{29}$$

where each $x_j$ is a decision variable. If the subset $\mathcal{S}_j$ is chosen, then $x_j = 1$; otherwise $x_j = 0$.

(2) Combinatorial Auction: consider a scenario where there are $n$ bids and $m$ available. Each item is associated with a subset $S_j \subseteq \{1, \cdots n\}$ representing the bidders interested in that particular item. The revenue generated by each bid $i$ is denoted as $b_i$. The Combinatorial Auction problem aims to allocate the bids to maximize the expected return.

$$\text{maximize} \quad \sum_{i=1}^{n} x_i b_i \tag{30}$$

$$\text{subject to} \quad \sum_{i \in S_j} x_i \leqslant 1 \quad \forall j \in \{1, \cdots, m\} \tag{31}$$

$$x_i \in \{0, 1\} \quad \forall i \in \{1, \cdots, n\} \tag{32}$$

where each $x_i$ is a decision variable. If bid $x_i$ is selected, then $x_i = 1$, otherwise $x_i = 0$.

(3) Capacitated Facility Location: assuming there are $m$ facilities and $n$ customers, the goal is to satisfy customers at a minimum cost. Let $f_i$ denote the cost of building facility $i \in \{0, \cdots, m\}$ and $c_{ij}$ denote the transportation cost of products from facility $i$ to customer $j \in \{0, \cdots, n\}$. The demand of customer $j$ is $d_j > 0$, and the capacity of facility $i$ is $u_i > 0$.

$$\text{minimize} \quad \sum_{i=1}^{m} \sum_{j=1}^{n} c_{ij} y_{ij} + \sum_{i=1}^{m} f_i x_i \tag{33}$$

$$\text{subject to} \quad \sum_{i=1}^{m} y_{ij} = 1 \quad \forall j \in \{1, \cdots, n\} \tag{34}$$

$$\sum_{j=1}^{n} d_i y_{ij} \leqslant u_i x_i \quad \forall i \in \{1, \cdots, m\} \tag{35}$$

$$y_{ij} \geqslant 0 \quad \forall i \in \{1, \cdots, m\}, \quad j \in \{1, \cdots, n\} \tag{36}$$

$$x_j \in \{0, 1\} \quad \forall j \in \{1, \cdots, n\} \tag{37}$$

where each Boolean variable $x_j$ and each continuous variable $y_{ij}$ are decision variables. If facility $j$ is built, $x_j = 1$, otherwise $x_j = 0$. In addition, variable $y_{ij}$ represents the percentage of demand $d_j$ that is assigned to facility $i$.

(4) Maximum Independent Set: consider an undirected graph $\mathcal{G} = (\mathcal{V}, \mathcal{E})$, wherein a subset $S \in \mathcal{V}$ of nodes is deemed an independent set when no edges interconnect any couple of nodes within $\mathcal{S}$. The task at hand is to determine the largest possible number of independent sets in the graph $\mathcal{G}$.

$$\text{maximize} \quad \sum_{v \in \mathcal{V}} x_v \tag{38}$$

$$\text{subject to} \quad x_u + x_v \leqslant 1 \quad \forall (u, v) \in \mathcal{E} \tag{39}$$

$$x_v \in \{0, 1\} \quad \forall v \in \mathcal{V} \tag{40}$$

where each Boolean variable $x_v$ is a decision variable. If node $v \in \mathcal{V}$ is selected in the independent set, then $x_v = 1$, otherwise, $x_v = 0$.

