# OpenReview forum: "CAMBranch: Contrastive Learning with Augmented MILPs for Branching"
_ICLR.cc/2024/Conference — ICLR 2024 poster_

### Official Review · Reviewer_EfKd · 2023-10-16

**Soundness:** 3 good
**Presentation:** 3 good
**Contribution:** 3 good
**Rating:** 6
**Confidence:** 5

**Summary:**

Update on November 19:

I raise my score to 6 based on the authors' responses. I am willing to keep discussing with the authors and the other reviewers to achieve a fully discussed final score.

---
This paper point out the shifting equivalence of the MILP problems. Then, the authors propose a contrastive learning approach based on the augmented MILP problems. Experiments demonstrate that CAMBranch achieves high performance with only 10% of the complete dataset.

**Strengths:**

1. Practical motivation. In practice, I found the similar problem that generating expert demonstration for industrial-level datasets is extremely time-consuming. Thus, the motivation of this paper is practical.
2. Clear writing. The paper is clearly structured and easy to go through flow.
3. Simple and effective approach. The idea of the shifting equivalence is simple and effective. Previous research uses GNN to tackle the symmetry in row and column orders. In this paper, the authors explicitly use the shifting equivalence via contrastive learning to enhance the training efficiency.

**Weaknesses:**

1. The illustration of experimental results requires to be improved. The bars missing detailed values on it. Compared with histograms, tables could be more compact for demonstration.
2. Missing comparisons to other ML approaches. Employing auxiliary tasks is an effective way to improve the training efficiency. Empirically, I found the simple auxiliary task employed in [1] can efficiently promote the training efficiency. Thus, can you provide more results comparing this data augmentation approach to other auxiliary tasks?
3. Marginal improvement in Figure 1. It seems the improvement of CAMBranch is not significant. Maybe a lighter GNN with fewer hidden layers and hidden nodes can achieve similar IL accuracy but requires less expert data.

**Questions:**

See weaknesses above.

---

> ### Author Response · Authors · 2023-11-14
> **Response to Reviewer EfKd**
>
> Thanks for your valuable insights and suggestions. Here are our responses:
>
> 1.
>
> Thanks for pointing this out. Actually, we have displayed the table-format results in Appendix Table 5. In the revised version, we will place Table 5 into the main text.
>
> 2.
>
> Maybe missing a bibliography [1]. Could you please add it? Thanks for your feedback!
>
> In Gasse et al. (2019), experiments results have demonstrated that GCNN outperforms other ML approaches by a large margin. Therefore, we did not include these ML approaches and CAMBranch will certainly outperform them now that CAMBranch’s performance is close to GCNN’s.
>
> Furthermore, it's essential to note that CAMBranch is a plug-and-play method, ensuring ease of implementation without conflicts with other auxiliary methods in most cases, and can be combined with other auxiliary methods. This adaptability and compatibility contribute to its versatility and potential for integration with other methods.
>
> 3.
>
> Currently, the model architecture of CAMBranch follows Gasse et al. (2019) and it only has ~10k parameters, light enough compared with those extremely deep graph neural networks like DeeperGCN. The hidden size is only 64 and extremely small for graph neural networks. The hidden layer is only one.
>
> To explore whether lighter GNN with less expert data can achieve similar IL accuracy or not, we conducted additional experiments. In our additional experiments, conducted with the same 10% training data (less expert data), we trained the GCNN while adjusting the hidden size to 32 and 16. Imitation results on Set Covering are shown in the following. The results highlight that merely reducing the hidden size leads to negligible performance changes. In contrast, our proposed CAMBranch significantly outperforms these approaches.
>
> |Model|acc@1|acc@5|acc@10|
> |---|---|---|---|
> |GCNN (10%, hidden size 16)|59.56 ± 0.70|82.75 ± 0.98|91.27 ± 0.67|
> |GCNN (10%, hidden size 32)|59.65 ± 0.30|83.36 ± 0.39|91.63 ± 0.22|
> |GCNN (10%, hidden size 64)|58.98 ± 0.69|82.97 ± 0.52|91.61 ± 0.44|
> |CAMBranch (10%, hidden size 64)|**65.27 ± 3.92**|**89.46 ± 2.98**|**96.14 ± 1.90**|
>
> Significantly, when considering MILP solving, while imitation learning (IL) accuracy holds importance, it isn't the primary metric of concern. In contrast, time, Nodes, and wins are the primary metrics and can actually measure the performance of a neural branching strategy. This discussion is detailed in the Appendix E Discussion section. It's crucial to note that while Strong Branching serves as the expert strategy for imitation learning, it might not represent the optimal branching strategy. Therefore, IL accuracy in imitating Strong Branching merely stands as an intermediate result and cannot fully demonstrate the models’ performance.
>
> The true essence lies in the metrics evaluated during the inference stage—solving instances fast, and generating fewer nodes. Figures 2, 3, and 4 display the results, demonstrating CAMBranch's superior performance in these crucial metrics, especially evident in handling significantly harder instances.

---

> > ### Comment · Reviewer_EfKd · 2023-11-18
> >
> > Thank you for pointing out the bibliography [1]. We have added it below. The simple auxiliary task employed in [1] is proposed to improve the learning accuracy of the proposed hybrid model. However, in practice, we found this auxiliary task may also improve the training efficiency with limited available data.
> >
> > Based on the results provided in your response, I have two further questions&suggestions:
> > 1. What about the training time of CAMBranch? As the main motivation of this paper is to reduce the time for data collection, I think the comparison on training overhead should also be considered together.
> > 2. Results on real-world MILP benchmarks are recommended. MILP instances from industry are usually more complex and challenging. Thus, the time-consuming data collection process on these benchmarks becomes the key bottleneck for ML-based optimization. Data collection on the four synthetic benchmarks is usually efficient. You might want to conduct experiments on real-world benchmarks like those in the NeurIPS ML4CO competition to further support your motivation.
> >
> > Anyway, the results in the paper and your response (both to me and to the other reviewers) are interesting and promising. Thus, I slightly raise my score to 6. I am willing to keep discussing with the authors and the other reviewers to achieve a fully discussed final score.
> >
> > [1] Gupta, Prateek, et al. "Hybrid models for learning to branch." Advances in neural information processing systems 33 (2020): 18087-18097.

---

> ### Author Response · Authors · 2023-11-19
> **Response to Reviewer EfKd**
>
> Thank you for your further suggestions. Regarding the comparison of training overhead, we acknowledge that CAMBranch requires additional computations compared to the original GCNN in Gasee et al. (2019). To this end, we evaluated the computational burden for both models, measured in terms of training time per batch. Below are the results of training on Set Covering with each batch containing 64 samples. From the results, we can see that, despite the introduced computation by CAMBranch, the observed difference in computational speed is deemed acceptable—merely a matter of several milliseconds. In summary, the training overhead associated with CAMBranch is within acceptable bounds.
>
> |Model|Training time per batch (s)|
> |---|---|
> |CAMBranch (10%)	|0.09666|
> |GCNN (10%)|	0.09216|
>
> We appreciate your second suggestion as well. Aligning with established works in this domain, such as Gasse et al. (2019) and Gupta et al. (2020), we presented the results of our proposed method on four classical datasets. These results underscore the potential of our approach. Then, as you pointed out, the industry presents more challenging MILPs with prolonged collection processes. In such scenarios, CAMBranch's augmentation component outpaces Strong Branching solving in generating labeled expert samples. Unlike the latter, CAMBranch removes the need to solve NP-hard instances, directly generating samples based on the collected small parts of data.
>
> Here, we first consider the Capacitated Facility Location Problem, which exhibits the lowest collection efficiency among the four MILP benchmarks in our paper. Generating 100k expert samples using Strong Branching to solve the instances takes 84.79 hours. In contrast, if obtaining the same quantity of expert samples, CAMBranch requires 8.48 hours (collecting 10k samples initially) plus 0.28 hours (generating the remaining 90k samples based on the initial 10k), totaling 8.76 hours—an **89.67% time savings.** The advantages will probably be more pronounced in more challenging benchmarks. Although our current results are based on the four classic benchmarks, the underlying idea of CAMBranch is promising and will promisingly offer insights to the community, which will also lead to further efforts for this community. We appreciate your valuable suggestion and will be committed to further exploration and contribution to our community based on your insights.

---

> > ### Comment · Reviewer_EfKd · 2023-11-19
> >
> > Thank you for the response and effort in addressing all of my comments. Most of them have been properly addressed. I will keep my current rating (6) and encourage the authors to further improve the paper according to my suggestions above.

---

### Official Review · Reviewer_8jS2 · 2023-11-01

**Soundness:** 3 good
**Presentation:** 3 good
**Contribution:** 2 fair
**Rating:** 5
**Confidence:** 3

**Summary:**

The authors propose **CAMBranch** an innovative approach that seeks to enhance the efficiency of Branch and Bound (B&B) algorithms for solving Mixed Integer Linear Programming (MILP) problems. Traditional B&B methods, while reliable, can be computationally intensive, motivating the exploration of machine learning to improve branching decisions. In particular, **CAMBranch** employs a machine learning framework based on contrastive learning and uses Augmented MILPs (AMILPs) to inform its branching strategies.

The key innovation of in the work lies in its utilization of AMILPs, which are equipped with expert decision labels, making them suitable for imitation learning. This approach aims to mimic the performance of Strong Branching, a highly effective but computationally expensive B&B policy. By leveraging the relationships between MILP and AMILP, the method extracts and utilizes node features from an augmented bipartite graph to inform its branching decisions.

The strengths include its novel application of contrastive learning in the domain of optimization and its potential to significantly reduce the computational burden associated with Strong Branching. The use of AMILPs for imitation learning could lead to more efficient and informed branching strategies, potentially improving the overall performance of B&B algorithms.

However, the success of the proposed algorithm hinges on the accuracy of the imitation learning model. If the model fails to capture the subtleties of Strong Branching, it could result in suboptimal decisions. Additionally, the integration of machine learning into B&B algorithms may introduce computational overhead, which could diminish the anticipated efficiency gains. Lastly, the generalizability of **CAMBranch** across diverse MILP problems remains to be thoroughly evaluated, as its effectiveness may vary depending on the problem's characteristics.

**Strengths:**

Strengths:

* Innovation in Branch and Bound (B&B) Methods: The paper discusses recent advancements in machine learning frameworks to enhance B&B branching policies for solving Mixed Integer Linear Programming (MILP), indicating a focus on innovation and improvement of existing methods.
* Potential for Improved Efficiency: If these machine learning-based methods can successfully imitate Strong Branching, they may offer more efficient alternatives to traditional B&B methods, which can be computationally intensive.

**Weaknesses:**

Potential weaknesses of CAMBRANCH:

1. **Fidelity of Imitation Learning:** The effectiveness  is predicated on the assumption that imitation learning can closely approximate the decisions made by Strong Branching. However, Strong Branching is known for its nuanced decision-making process, which considers a multitude of factors and potential future states. Replicating this complexity through imitation learning is challenging. If the learning model fails to encapsulate the depth of Strong Branching's strategy, the resultant branching decisions could be significantly less efficient, negating the primary advantage of CAMBRANCH.

2. **Computational Overhead of Machine Learning Integration:** While machine learning models offer the promise of improved decision-making, they also introduce computational overhead. Training models, especially those based on contrastive learning, require significant computational resources. Furthermore, the real-time application of these models within the iterative B&B process could lead to increased computational demands, potentially offsetting the efficiency gains from more informed branching.

3. **Sensitivity to Hyperparameter Tuning:** Machine learning models, particularly those used in imitation learning, are sensitive to hyperparameter settings. The performance of CAMBRANCH could be highly dependent on the choice of learning rate, batch size, and other hyperparameters. Finding the optimal configuration can be a time-consuming process that requires extensive experimentation and computational resources.

4. **Generalizability and Robustness Concerns:** The diversity of MILP problems poses a significant challenge to the generalizability of CAMBRANCH. Different MILP instances can vary drastically in terms of size, structure, and complexity. CAMBRANCH must demonstrate robust performance across a wide array of problems to be considered a viable alternative to existing B&B methods. Additionally, the model's robustness to adversarial inputs or unusual problem structures remains to be thoroughly evaluated.

In conclusion, while CAMBRANCH presents an innovative approach to improving B&B algorithms for MILP problems, its success is contingent upon overcoming the intricate challenges associated with imitation learning, computational efficiency, hyperparameter sensitivity, and the robustness of its application across diverse problem sets.

**Questions:**

1. **How does the model's architecture influence the quality of imitation learning?**
   - The architecture of the neural network used for imitation learning plays a crucial role in its ability to capture the decision-making process of Strong Branching. How does the choice of architecture impact the model's ability to generalize across diverse MILP instances?

2. **What is the impact of contrastive learning's positive and negative sample selection on the performance?**
   - In contrastive learning, the selection of positive and negative samples is critical for learning meaningful representations. How does CAMBranch ensure the selection of informative positive and negative samples during training? Could the incorporation of hard negative mining or other advanced sampling strategies improve the model's performance?

3. **How does the algorithm address the exploration-exploitation trade-off during the B&B process?**
   - The Branch and Bound algorithm involves a delicate balance between exploring new branches and exploiting known promising paths. How does CAMBranch navigate this trade-off? Are there mechanisms in place to prevent premature convergence on suboptimal branches or to encourage exploration when necessary?

4. **How does CAMBranch handle the interpretability and explainability of its branching decisions?**

---

> ### Author Response · Authors · 2023-11-14
> **Response to Reviewer 8jS2**
>
> Thanks for your valuable feedback. Maybe there are misunderstanding and we will clarify it in the following responses.
>
>  **Response to the questions**
> 1.
>
> Thanks for pointing this out. There may be some misunderstanding for *"generalize across diverse MILP instances"*. In our work, we follow Gasse et al. (2019) and focus on homogeneous MILP solving, i.e., the training and test MILP instances are the same type of problems, which is a popular setting in industry practice. Such models do not need to generalize across diverse MILP instances in the homogeneous setting but need to generalize across the same type of instances, i.e., those with different coefficients and different scales of variables and constraints.
>
> Within this specialized context, we prefer to choose light graph neural networks due to their high efficiency during the inference phase. Following Gasse et al. (2019), we leverage the GCNN as CAMBranch’s backbone model. GCNN can capture the features of MILP bipartite graphs and exhibits robust learning capabilities from expert samples during the imitation learning phases.
>
> 2.
>
> In CAMBranch, as detailed in our paper, “we leverage this principle by viewing a MILP and its corresponding AMILP as positive pairs while considering the MILP and other AMILPs within the same batch as negative pairs.” This follows the simplest way of negative sampling strategy in contrastive learning, and our results already demonstrate its contribution to superior performance. Moreover, incorporating more advanced hard negative sampling strategies is a choice to further improve the model’s performance.
>
> 3.
>
> Since the CAMBranch pipeline operates in two stages, the exploration-exploitation trade-off isn't our concern. Specifically, we first collect expert samples for imitation learning, followed by the training of the policy model after this collection phase. Once the model is trained, it becomes directly available for inference without the necessity to address mechanisms preventing premature convergence.
>
> 4.
>
> Given that our CAMBranch methodology imitates Strong Branching, it aligns with Strong Branching decisions in most cases. The interpretability of neural branching remains an ongoing research focus and an open problem in this field, and we intend to explore this further in our future work.
>
> **Response to the weaknesses**
>
> 1. As depicted in the results presented by Gasse et al. (2019), the imitation learning paradigm demonstrates remarkable superiority in this domain. Since CAMBranch follows Gasse et al. (2019), CAMBranch also demonstrates superior performance. Therefore, there is no such concern.
>
> 2. Indeed, our CAMBranch model demands modest computational resources during training phases. For the inference phases, it is the same as the models in Gasse et al. (2019), both of which show faster solving speed, especially in hard instances.
>
> 3. Since our CAMBranch follows Gasse et al. (2019) and the model backbone are extremely light graph neural networks, grid search for hyperparameter tuning does not need too many resouces. Actually, both CAMBranch and Gasse et al.(2019) are not sensitive to the hyperparameter tuning, further streamlining the optimization process.
>
> 4. As responding in Question 1, “we focus on homogeneous MILP solving, i.e., the training and test MILP instances are the same type, which is a popular setting in industry practice.” From the results in our paper, CAMBranch as well as GCNN in Gasse et al. (2019) obviously generalize well in homogeneous MILP instances. Therefore, no such concerns.

---

> ### Author Response · Authors · 2023-11-21
> **Response to Reviewer 8jS2**
>
> Dear Reviewer 8jS2,
>
> We have revised our paper according to the other three reviewers' suggestions.
>
> We would appreciate it if you could confirm that our responses address your concerns. We are happy to answer more if you have any remaining concerns or questions.
>
> Best,
>
> Authors

---

### Official Review · Reviewer_mGEq · 2023-11-02

**Soundness:** 3 good
**Presentation:** 2 fair
**Contribution:** 3 good
**Rating:** 6
**Confidence:** 4

**Summary:**

This paper proposes CAMBranch, which generates augmented MILPs with shifting variables and identical variable selection decisions, and uses contrastive learning ot use the augmented MILPs for learning to branch. Experiments demonstrate that CAMBranch can effectively improve the sample efficiency.

**Strengths:**

1. This paper identifies the challenge of  sampling time in Learn2Branch, which is meaningful.
2. The augmentation strategy has a theoretical gurantee and is insightful.

**Weaknesses:**

1. CAMBranch does not perform well on relatively easy datasets. The improvement is not significant compared with GCNN baseline on many medium and hard datasets. Fow example, results in Table 8 do not reveal that CAMBranch consistently outperforms GCNN.
2. The augmentation method is tailored for the branching task, while not easy to be transfered to other algorithms in B & B solvers. Therefore, the application is limited.
3. Which sample ratio, with CAMBranch, leads to a comparative results with the model trained with the full dataset? What if use the full training dataset  with this augmentation? Will it stll bring improvement?
4. How many instances does it need to generate 20k expert samples? The 10% means using 10% MILP instances or 10% expert samples? How many expert samples can one instance produce? Experiments should be conduct to investigate the effect of the number of MILP instances and the expert samples. Also, the accumulative time used in collecting the expert samples should be reported.

**Questions:**

1. Sometimes GCNN (10%) even outperforms GCNN (e.g., MIS in Figure 2). Why?
2. See weaknesses.

---

> ### Author Response · Authors · 2023-11-14
> **Response to Reviewer mGEq**
>
> Thanks for your valuable feedback. CAMBranch aims to mitigate issues of the extremely long collection time of expert samples by innovatively generating new expert samples from existing ones. Notably, CAMBranch leverages only a small fraction of the expert samples, compared to the GCNN method in Gasse et al., (2019). Its adaptable nature makes it a plug-and-play solution, seamlessly integrated into various neural branching methods for B&B applications.
>
> **Response to the weaknesses:**
> 1.
>
> Thanks for highlighting this. While CAMBranch might not consistently outperform GCNN in a small subset of easy and medium instances, it's important to note that the differences are marginal in these scenarios. However, the true strength of CAMBranch shines through in harder instances.
>
> For instance, in Table 5, which is the primary focus of this paper, the Combinatorial Auction Problem at the medium level reveals that CAMBranch performs with a mere 0.3s difference compared to GCNN(10%), specifically 12.68s versus 12.38s. This trend is consistent across most cases where CAMBranch doesn't outperform GCNN (10%), as similarly observed in Table 8.
>
> However, when tackling genuinely challenging instances, especially in Set Covering, Capacitated Facility Location, and Maximum Independent Set (problems significantly more complex than the hard Combinatorial Auction Problem, evident from the solving times and the number of instances successfully solved), CAMBranch distinctly demonstrates its superiority in solving these intricate problems. For instance, in Set Covering, CAMBranch has a solving time of 1427.02s compared to GCNN 10%'s 2385.23s. This exemplifies the practical utility of CAMBranch since real-world instances are rarely as simple as the Combinatorial Auction Problem; most instances pose challenges of much higher complexity, such as hard Set Covering problems.
>
> Taking this into account, CAMBranch serves as a robust, plug-and-play solution, especially in scenarios where inefficient data collection hampers performance. It consistently exhibits high efficacy in MILP problem-solving while requiring fewer resources, making it a compelling choice for improving methodologies in such settings.
>
> 2.
>
> CAMBranch, its plug-and-play nature facilitates easy implementation, specifically enhancing the neural branching decision part in the B&B algorithm. While the primary focus of CAMBranch is on the branching part, it's worth noting that the core idea of generating expert samples sharing identical decisions from the original ones holds potential insights for other decision parts within B&B solvers, such as node selection and cutting planes. The fundamental idea of generating new labeled expert samples from original expert decisions holds promise for broader applicability within the B&B framework.
>
> 3.
>
> The CAMBranch method is proposed to alleviate challenges arising from inefficient expert sampling.  Therefore, in such contexts, it is more meaningful to use CAMBranch when facing data-scarce issues. While CAMBranch is primarily tailored to address data scarcity issues, its adaptability allows users to combine this methodology with a full training dataset. This integration aids in the form of data augmentation, further fostering improvements in the training process.
>
> 4.
>
> Thanks for pointing these out. In the introduction part, we report that collecting 100k expert samples for four types of problems requires 26.65 hours, 12.48 hours, 84.79 hours, and 53.45 hours, respectively. Therefore, collecting 20k experts（20% of the whole data) needs about 20% of the former time periods.
>
> The 10% means 10% expert samples, not MILP instances. Generating MILP instances is trivial, but acquiring the expert samples is challenging since numerous NP-hard MILP instances need to be solved, making it a time-consuming endeavor.
>
> Additionally, it's worth noting that Gasse et al. (2019) extensively explored the relationship between the number of solved MILP instances and expert samples in their Appendix Section 1. Our proposed methods have nearly identical observations to Gasse et al. (2019).
>
> **Response to the question:**
>
> Yes, it is an interesting observation and we've thoroughly investigated this observation through multiple experiments, consistently confirming its occurrence. One potential reason is that Strong Branching is not always the optimal expert strategy for branching and sometimes may lead to sub-optimal decisions, even if it makes high-quality decisions most of the time. This aligns with similar observations outlined in recent studies by Scavuzzo et al. (2022), Dey et al. (2023), and Gamrath et al. (2020). Refer to our Appendix Discussion part, “Recent studies by Scavuzzo (2022), Dey (2023), and Gamrath (2020) have even suggested that Strong Branching may underperform in certain cases, falling short of problem-specific rules.” This intriguing observation inspires us to delve into this phenomenon, for more exploration in future work.

---

> > ### Comment · Reviewer_mGEq · 2023-11-20
> >
> > Thanks for the authors' response. My comments are as below.
> > 1. It would be better if the authors could provide further insights on why CAMBranch performs better on genuinely challenging instances than on easy instances.
> > 1. I still think the main contribution is limited in the branching task. The idea of generating new labeled expert samples is not novel, and what matters is the augmenting strategy.
> > 1. What I really challenge in Weakness 3 and 4 are as follows. When we have a collection of MILP instances, we have to determine whether we should solve all of them to obtain expert samples or solve parts of them and use CAMBranch to obtain augmented ones. Here is a trade-off between the efficiency and the performance. So I think the authors should provide more quantitative information on in which cases we should use CAMBranch. For example, futhter analysis on the relationship between the performance and the number of MILP instances, expert samples, and the number of augmented samples. If the authors can provide this, I would like to raise my score.

---

> > > ### Author Response · Authors · 2023-11-22
> > > **Response to Reviewer mGEq**
> > >
> > > Thanks for your further feedback. We appreciate your insightful suggestions regarding the analysis of the relationship between performance and various factors such as the number of instances used for sample collection and the number of expert samples. Following your suggestions, we conducted the following experiments and present the findings from the table on Combinatorial Auction Problem:
> > >
> > > (1) **Impact of Expert Sample Size:** From pairs (I, II), (III, IV), and (V, VI), we delve into the relationship between performance and the number of expert samples, holding the number of MILP instances constant. The results indicate that increasing the number of expert samples generally enhances performance, as evident in pairs (I, II) and (III, IV). However, in scenarios with a relatively small number of instances, such as (V, VI), augmenting expert sample size may not necessarily lead to a performance increase.
> > >
> > > (2) **Influence of MILP Instance Count:** Examining pairs (I, III, IV) and (II, IV, VI), we further explore the relationship between performance and the number of MILP instances, maintaining a consistent expert sample size. The observations suggest a general trend of performance improvement with an increase in the number of MILP instances.
> > >
> > > |Exp ID|	Model|	Time(Easy)|	Nodes(Easy)|	Time(Medium)|	Nodes(Medium)|	Time(Hard)|	Nodes(Hard)|
> > > |---|---|---|---|---|---|---|---|
> > > |I|	CAMBranch 362 instances 10k expert samples| 2.03|	91|	12.68|	758|	131.79|	9074|
> > > |II|	CAMBranch 388 instances 20k expert samples|1.68|	91|	9.92|	762|	103.38|	9050|
> > > |III|	CAMBranch 100 instances 10k expert samples |1.67|	94|	10.02|	757|	111.52|	9242|
> > > |IV|	CAMBranch 100 instances 20k expert samples |1.64|	90|	10.03|	751|	109.58|	9195|
> > > |V|	CAMBranch 50 instances 10k expert samples| 3.25|	98|	20.58|	796|	158.24|	9834|
> > > |VI|	CAMBranch 50 instances 20k expert samples|3.18|	97|	20.59|	819|	166.02|	10892|
> > >
> > > Based on these observations, to answer the question of when to use CAMBranch, our initial conclusion is that CAMBranch demonstrates more potential when there is sufficient number of MILP instances, but not necessarily too large scale (Using several hundreds of MILP instances can already achieve superior performance). Moreover, increasing the number of expert samples provides benefits for performance, as supported by our empirical findings.
> > >
> > > Furthermore, the number of augmented samples is determined by the epoch at which each CAMBranch model achieves its best performance on the validation set. During each training step, an equivalent quantity of augmented samples to those in each batch is generated simultaneously for contrastive learning. These augmented samples are used only once in each training step. Upon completion of the training process, the total number of augmented samples for the saved best model depends on the epoch at which the best model was achieved. In such contexts, here we report the number of one-time augmented instances for each model: I (670k), II (440k), III (550k), IV (720k), V (250k), and VI (380k).
> > >
> > > For deeper insights into the superior performance of CAMBranch in genuinely challenging instances, we offer a potential explanation here. Instance-solving performance hinges significantly upon its branching decisions’ quality (Achterberg and Wunderling, 2013). Notably, suboptimal branching decisions can exponentially escalate the computational workload, especially on hard instances, significantly prolonging the solving process. In contrast, high-quality decisions can benefit solving and lead to a reduced number of nodes. Our CAMBranch is trained with various expert samples, enabling it to generate superior branching decisions. Thus, in challenging instances where the quality of decisions makes a pronounced difference, CAMBranch excels. This advantage is less apparent in easy and medium instances where the impact of branching decisions is not as discernible.
> > >
> > > **References:**
> > >
> > > Achterberg, T., & Wunderling, R. (2013). Mixed integer programming: Analyzing 12 years of progress. In *Facets of combinatorial optimization: Festschrift for martin grötschel* (pp. 449-481). Berlin, Heidelberg: Springer Berlin Heidelberg.

---

> > > > ### Comment · Reviewer_mGEq · 2023-11-22
> > > >
> > > > Thanks for the authors' responses. I believe the additional experiments will provide meaningful insights. I have raised my score from  5 to 6. Since these experiments are only conducted on one datasets, I expect further results on more datasets and more hyperparameters.

---

### Official Review · Reviewer_vbTP · 2023-11-15

**Soundness:** 3 good
**Presentation:** 3 good
**Contribution:** 3 good
**Rating:** 6
**Confidence:** 3

**Summary:**

The paper proposes a data-efficient imitation learning algorithm for learning a branching policy in B&B algorithm.
The proposed method augments demonstration data of Strong Branching decisions using mathematically complete rule and introduces contrastive learning by using the augmented data as positive samples.
Experiment results demonstrate that the proposed method can significantly outperform existing imitation learning-based MILP solvers.

**Strengths:**

The proposed method of augmentation rule and contrastive learning idea is well-motivated, concise and effective.
The performance improvement from previous method in low-data situations is significant.

**Weaknesses:**

The presentation of main result should be improved.
- CAMBranch (100%) result should be discussed. Despite this work aiming in data scarce situation, understanding how it behaves with the full dataset is critical information, considering the results with 10% of data is not outperforming the baseline using full data.
Even if CAMBranch using the full data can't outperform the baseline using full data, it is still beneficial to share this limitation with the community if proper discussion is provided.
However, without the result, I am negative about introducing this method to the community as a MILP solver because it is unknown whether it limits the potential of imitation learning-based MILP methods.
- The main results appear to be promising but could be presented better. Neither Figure 2-4 nor Table 5 looks optimal in their current forms.
- The result of Table 8 deserves to be in main text, considering the goal of this work. Also, evaluation in another domain with more dramatic performance gap could be better for the comparison. The presentation also has room for improvement; it is hard to compare the metrics of models using the same amount of data.

If this concern is resolved I am willing to vote for accepting this paper.
However, as it seems that a major revision is required, including the core results of the paper, I also agree if other reviewers or AC recommend submitting this work to next venue.

**Questions:**

What would be the intuition behind using shifted geometric mean for evaluation?  And what is the $s$ value for each metric?

---

> ### Author Response · Authors · 2023-11-15
> **Response to Reviewer vbTP**
>
> Thanks for your valuable feedback and suggestions. The inspiration behind the CAMBranch project stems from real-world challenges encountered in the industry, i.e., the low sample efficiency observed during the collection of expert samples for imitation learning. To mitigate this, CAMBranch introduces a solution by generating labeled expert samples from the original datasets. This approach alleviates the need to collect a large quantity of expert samples. With a relatively small amount of data, we achieve superior and acceptable performance. The implications of this approach are highly meaningful in real-world applications.
>
> **Response to the weaknesses:**
>
> 1.
>
> Thanks for pointing this out. Your concern is that “*it is unknown whether it limits the potential of imitation learning-based MILP methods.*” While we understand that discussing CAMBranch (100%) compared to GCNN (100%) is one approach to alleviate this concern directly, we would like to propose an alternative way, i.e., to compare CAMBranch (10%) and GCNN (10%) since you can **assume the full datasets have 10k expert samples** which are also enough for training usually and the performance for GCNN (10%) is still acceptable in most of the cases. Therefore, after comparing GCNN (10%) and CAMBranch (10%), we can observe that CAMBranch (10%) shows high superiority. Thus, this observation suggests that CAMBranch will probably not limit the potential of imitation learning-based methods. Experiments for CAMBranch (100%) are also as the following. Here for combinatorial auction problem. The same observations are obtained.
>
> Easy
> |Model|Time|Wins|Nodes|
> |---|---|---|---|
> GCNN (100%)	|1.96	|4/100|	87
> GCNN (10%)	|1.99	|2/100|	102
> CAMBranch (10%)	|2.03|	1/100|	91
> CAMBranch (100%)|	1.73|	93/100|	88
>
> Medium
> |Model|Time|Wins|Nodes|
> |---|---|---|---|
> GCNN (100%)	|11.30|	7/100|	695
> GCNN (10%)|	12.38|	3/100|	787
> CAMBranch (10%)|	12.68|	2/100|	758
> CAMBranch (100%)|	10.04|	88/100|	690
>
> Hard
> |Model|Time|Wins|Nodes|
> |---|---|---|---|
> GCNN (100%)|	158.81|	4/94|	12089
> GCNN (10%)|	144.40|	2/100|	10031
> CAMBranch (10%)|	131.79|	11/100|	9074
> CAMBranch (100%)|	109.96|	83/100|	8260
>
> In fact, In practical scenarios, it is hard to measure the exact number of samples required to obtain an acceptable policy model. For some problems, maybe 10k is enough, while for others 100k, 200k, or more will lead to a better performance. In such contexts, CAMBranch is proposed to ensure that with any available dataset, one can use CAMBranch’s augmented approach to train a powerful policy model. Notably, CAMBranch initiates the process with **data augmentation**, enabling the generation of **Hundreds of Thousands of labeled samples** that are totally different data one another for neural networks. CAMBranch significantly mitigates the time-intensive data collection process by providing an abundance of augmented expert samples. The results demonstrate CAMBranch’s superiority and highlight its value.
>
> 2.
>
> Table 5 follows the paper by Gasse et al. (2019), a format also employed by other papers within this domain. We have presented Figures 2-4 as a visual representation of the data in Table 5, as we believe it provides a clearer comparison of the gaps between different methods in different difficulty levels. We are open to exploring alternative methods to present the results and welcome any suggestions you may have. Thank you for your input.
>
> 3.
>
> Thank you for your advice. While we aspire to include Table 8 in the main text, space constraints have compelled us to make the paper as concise as possible. Given the abundance of intriguing observations that warrant discussion, we have chosen to place Table 8 in the Appendix to ensure a comprehensive exploration.
>
> **Response to the question:**
>
> Utilizing the shifted geometric mean for metric calculations stands as a widely accepted practice in MILP benchmarking within the community, which is also employed by Gasse et al. (2019) and Zarpellon et al. (2021), two representative papers in this domain. Shifted geometric means have the advantage of neither being compromised by very large outliers (in contrast to arithmetic means) nor by very small outliers (in contrast to geometric means).
>
> The solving time for different instances, even the same instance solved with different seeds, inherently has the potential for substantial variations due to the NP-hard nature. Therefore, the community use shifted geometric mean to facilitate a more equitable comparison. In such contexts, $s$ in our paper is set to 1 for time and 100 for Nodes, following the previous work.
>
>
> **References**:
>
> Gasse, M., Chételat, D., Ferroni, N., Charlin, L., & Lodi, A. (2019). Exact combinatorial optimization with graph convolutional neural networks, NIPS.
>
> Zarpellon, G., Jo, J., Lodi, A., & Bengio, Y. (2021, May). Parameterizing branch-and-bound search trees to learn branching policies. AAAI.

---

> > ### Comment · Reviewer_vbTP · 2023-11-20
> > **Response**
> >
> > Thank you for the author's response.
> >
> > The author's response addressed the most important concern I had, including the crucial experimental result.
> > I would vote to accept this work when those modifications are well-reflected in the revised version.
> >
> > Some responses to the author's comments:
> >
> > 2. I apologize for the vague feedback.
> > As proposed by the reviewer EfKd, Table 5 seems to be a better format if presented with some effort on readability (e.g., bold, different line styles).
> > Alternatively, the main text can include only one difficulty for the domain (e.g., hard) and provide results on other difficulties in the appendix unless the authors have any important argument on comparison results tendency over different difficulties.
> > In this case, the authors may be able to keep the graph format while showing comparisons of different metrics (i.e., time, node, and win) in one row.
> > This is a suggestion and may not be the optimal format as well, but I believe the current presentation should be improved.
> >
> > 3. I also think the main results in Figure 2-4 should be prioritized over the results in Table 8.
> > I suggest, but do not insist, moving Table 8 to the main text, especially if the authors make the main results presentation more compact (as proposed in 2. in this comment) and have more space in the paper.
> > Still, as mentioned in the first review, Table 8 results can be improved by changing the evaluation domain and better presenting the results.

---

> > > ### Author Response · Authors · 2023-11-20
> > > **Response to Reviewer vbTP**
> > >
> > > Thanks for your valuable comments. We have revised our paper and uploaded it, according to your and Reviewer EfKd’s suggestions. For the result presentation, we finally chose to follow Gasse et al. (2019). Moreover, we added more analysis about measuring data collection efficiency and training overhead comparison in the main text, which are suggestions from Reviewer EfKd. Also, we added a part about evaluating CAMBranch on full datasets, which are from you and Reviewer mGEq. These two parts further verify the effectiveness and efficiency of our proposed method.
> > >
> > > We are continuously improving our paper. Thanks for your suggestions again!

---

> > > ### Author Response · Authors · 2023-11-21
> > > **Improve Table 8 by experiments on another domain with a more dramatic performance gap**
> > >
> > > Following your suggestions, we have conducted experiments on another domain with a more dramatic performance gap to improve Table 8. You can see the complete Table in the revised supplementary material Table 10.  Here we list the results on Hard level instances of Maximum Independent Set, which has an obvious performance gap.
> > >
> > >
> > > |Model|	Time|	Wins|	Nodes|
> > > |---|---|---|---|
> > > |GCNN (5%)|	607.48|	2/39|	16850|
> > > |GCNN (10%)|	695.96|	0/20|	17034|
> > > |GCNN (20%)|	465.00|	16/43|	12998|
> > > |CAMBranch (5%)|	592.81|	1/41|	15480|
> > > |CAMBranch (10%)|	496.86|	1/40|	10828|
> > > |CAMBranch (20%)|	**416.24**|	**20/49**|	**10455**|
> > >
> > > The results demonstrate a consistent upward trend in performance across multiple metrics, including solving time, number of wins, and nodes, with increasing sample size. Notably, CAMBranch (20%) achieved the highest number of solved instances in the "Hard" level, 49 instances in total, underscoring its superiority. Moreover, the comparison between the two methods reveals that CAMBranch consistently outperforms GCNN across all three ratios, emphasizing the superior capabilities of our proposed framework.
> > >
> > > This additional experiment further reinforces the effectiveness of our proposed methods. Thank you again for your valuable suggestions.

---

### Author Response · Authors · 2023-11-22
**General Response**

We thank all the reviewers for their suggestions and appreciation of our work. The revised version of our paper and supplementary material have been successfully uploaded. In the main text, here are the changes:
* For the result presentation, we finally chose to follow Gasse et al. (2019) to place a result table in the main text.
* We added more analysis about measuring data collection efficiency and training overhead comparison in the main text.
* Also, we added a part about evaluating CAMBranch on full datasets.

More experiments are running following the reviewers’ suggestions and the results will be added to the appendix if time permits. We are continuously improving our paper. Thanks for all the reviewers’ efforts!

---

### Meta-Review · Area_Chair_tjoT · 2023-12-07

**Metareview:**

The paper introduces CAMBranch, a method that enhances branching policies in Mixed Integer Linear Programming (MILP) by integrating contrastive learning with Augmented MILPs (AMILPs). CAMBranch aims to improve sample efficiency in training machine learning models for branching decisions, leveraging limited expert data. The method shows significant performance improvements, especially in data-scarce situations, by generating a large number of labeled expert samples from original datasets. Main strengths include interesting use of contrastive learning and data augmentation in MILP branching strategies, performance improvement in low-data scenarios.

Weaknesses:
- Underperforms on easier datasets and shows limited improvement on many medium and hard datasets.
- Specific to branching tasks in B&B solvers, potentially limiting broader applicability.
- The paper lacks comparisons with other machine learning approaches (other than GCNN) and real-world MILP benchmarks.

**Justification For Why Not Higher Score:**

The specificity to B&B tasks limits the broader applicability of this work.

**Justification For Why Not Lower Score:**

All reviewers argued the paper is marginally above acceptance threshold. I believe this paper can be accepted.

---

### Decision · Program_Chairs · 2024-01-16

Accept (poster)